# PD-L2 controls peripherally induced regulatory T cells by maintaining metabolic activity and Foxp3 stability

Benjamin P. Hurrell[1], Doumet Georges Helou[1], Emily Howard[1], Jacob D. Painter[1], Pedram Shafiei-Jahani[1], Arlene H. Sharpe [2] & Omid Akbari [1] ✉

Regulatory T (Treg) cells are central to limit immune responses to allergens. Here we show that PD-L2 deficiency prevents the induction of tolerance to ovalbumin and control of airway hyperreactivity, in particular by limiting pTreg numbers and function. In vitro, PD-1/PD-L2 interactions increase iTreg numbers and stability. In mice lacking PD-L2 we find lower numbers of splenic pTregs at steady state, producing less IL-10 upon activation and with reduced suppressive activity. Remarkably, the numbers of splenic pTregs are restored by adoptively transferring PD-L2[high] dendritic cells to PD-L2[KO] mice. Functionally, activated pTregs lacking PD-L2 show lower Foxp3 expression, higher methylation of the Treg-Specific Demethylation Region (TSDR) and a decreased Tricarboxylic Acid (TCA) cycle associated with a defect in mitochondrial function and ATP production. Consequently, pyruvate treatment of PD-L2[KO] mice partially restores IL-10 production and airway tolerance. Together, our study highlights the importance of the PD-1/PD-L2 axis in the control of metabolic pathways regulating pTreg Foxp3 stability and suppressive functions, opening up avenues to further improve mucosal immunotherapy.

Respiratory tolerance is a process by which the immune system becomes hyporesponsive to inhaled innocuous environmental antigens. This prevents the inappropriate activation of CD4[+] T helper type (Th) 2 cells and the production of inflammatory cytokines including interleukin (IL)−4, IL-5, IL-9, and IL-13, which can lead to collateral tissue damage at the site of inflammation such as the lungs[1]. One mechanism by which excessive reactivity is regulated is through the induction of regulatory T cells (Tregs)[2]. Tregs express the key transcription factor Forkhead box protein 3 (Foxp3), which is central for suppressive function and production of immune regulatory molecules such as anti-inflammatory cytokines IL-10 and tumor growth factor Beta (TGF-β)[3]. Signaling via interleukin-2 receptor (CD25) is crucial for Treg cell identity and function, as partial loss of CD25 was shown to interfere with the maintenance, heterogeneity, and suppressive function of the Treg cell pool, particularly in inflammatory environments[4]. The majority of peripheral Tregs originate from the thymus (tTreg),

but they may also be induced in the periphery from naive CD4[+] T cells (pTreg) or derived in vitro with TGF-β and IL-2 from naive T cells (iTreg)[5–7], as tTregs are distinguished from pTregs and iTregs by their expression of glycoprotein neuropilin-1 (Nrp1) and transcription factor Helios[8]. Importantly, induction of Tregs in the periphery was shown to inhibit the proliferation and activity of CD4[+] and CD8[+] T cells, as dysfunctional Treg activity is associated with a plethora of immune disorders including allergies[9].

Co-inhibitory receptors control immune reactions by delivering inhibitory signals to activated T cells to avoid self-damage[10]. The last decades of research in Immunology saw a particular focus on co-inhibitory signals, since cancer cells are able to hijack these inhibitory pathways, causing T cell exhaustion and decreased T cell-mediated immune responses against tumors[11]. Programmed cell death protein (PD)−1 is one of the main inhibitory checkpoints involved in T-cell inhibition, as it is rapidly induced following activation of T cells in

[1]Department of Molecular Microbiology and Immunology, Keck School of Medicine, University of Southern California, Los Angeles, CA, USA. [2]Department of Immunology, Harvard Medical School, Boston, MA, USA. ✉e-mail: akbari@usc.edu

response to various stimuli[12]. However, studies have shown that in the context of chronic antigen stimulation, PD-1 expression is sustained and associated with T cell exhaustion and progression of disease[13]. PD-1 binds not only to PD-L1 which is constitutively expressed by antigen-presenting cells and upregulated by interferons, but also to PD-L2, which is mainly inducible[14–16]. Binding of PD-L1/2 to PD-1 induces phosphorylation of two tyrosine motifs in its cytoplasmic tail, the immunoreceptor tyrosine-based inhibitory motif and the immunoreceptor tyrosine-based switch motif, regulating interactions with Shp-2, Zap-70, and Lck for PD-1-mediated inhibitory functions[14]. A growing number of cancers are treated with FDA-approved anti-PD-1 or anti-PD-L1 antibodies blocking PD-1 or PD-L1 with response rates for PD-1/PD-L1 blockade in approved indications ranging from 13 to 69% depending on tumor type[11,17]. In light of this success, a better understanding of the PD-1/PD-L1/2 axis in various cells and diseases will therefore help in the design of novel therapeutic approaches.

Naive CD4+ T cells that enter peripheral circulation can be induced to express Foxp3 through the engagement of both the TCR and PD-1 on the surface[18]. Furthermore, activated Tregs express higher levels of PD-1 that contribute to their activation, as blocking PD-1 or loss of PD-L1 leads to enhanced Treg cell activity[19,20]. However, both PD-L1 and PD-L2 are involved in respiratory tolerance and even if their functions may overlap, they can control T cell responses in a largely context-dependent manner. For instance, whereas PD-L1 is upregulated by TLR4 and STAT1 signaling, PD-L2 expression depends on IL-4Rα and STAT6, together suggesting that a PD-L1 or PD-L2 dominance may be explained by the local microenvironment[12]. Furthermore, studies show that PD-L2 binds to PD-1 with threefold stronger affinity compared to PD-L1, highlighting the significance of PD-L2 in the formation of the PD-1-mediated induction of immune suppression[21]. In line with this, we previously found that the severity of airway inflammation in response to ovalbumin (OVA) is higher in PD-L2[KO] mice as compared to PD-L1[KO] mice, suggesting that PD-L2 plays a major role in airway inflammation[16]. Although PD-L1 was shown to be required for the development, maintenance, and function of pTregs[18], the role of PD-1/PD-L2 in this context is not fully understood and may be critically important for the induction of respiratory tolerance.

Herein we show that PD-L2 deficiency prevents the induction of respiratory tolerance and control of airway hyperreactivity (AHR). In vitro and in vivo results show that PD-1/PD-L2 interactions result in an increase in pTreg numbers, Foxp3 expression, TCA cycle, and mitochondrial function as well as lower TSDR methylation as compared to controls. Under homeostatic conditions, mice lacking PD-L2 have lower numbers of pTregs, as adoptive transfer of PD-L2[high] dendritic cells restores these numbers. Finally, pyruvate treatment of pTregs in mice lacking PD-L2 partially restores pTreg function and respiratory tolerance. Our findings, therefore, highlight the importance of PD-L2 in maintaining pTreg metabolic activity and Foxp3 stability, together contributing to the development of immunoregulatory pTregs and respiratory tolerance.

## Results

### PD-L2 is induced in the lungs following OVA airway challenge
PD-L1 and PD-L2, the two ligands of PD-1, are known to have different expression dynamics during inflammation in the lungs[14]. We first challenged WT BALB/c mice intranasally (i.n.) with PBS or OVA on 3 consecutives days and on day 4 measured the expression of PD-L2 in total lung CD45+ cells. Compared to controls, we first found low PD-L2 expression in PBS-treated mice compared to staining control (Fig. 1A). Remarkably, however, OVA challenge efficiently induced the frequency of PD-L2 expression in total lung CD45+ cells, representing overall 13-fold induction in the numbers of lung CD45+ cells compared to controls (Fig. 1A, B). This is in sharp contrast with PD-L1 expression patterns, as we observed that PD-L1 is already highly expressed in PBS-treated lung CD45+ cells and only induced by 2-fold following OVA intranasal

challenge (Supplementary Fig. 1A). We further found that CD11c+MHCII+ cells strongly upregulated PD-L2 expression as compared to PD-L1 in response to OVA challenge, representing the majority of CD45+ PD-L2-expressing cells in the lungs (Fig. 1B and Supplementary Fig. 1B, C).

### PD-L2 deficiency prevents the induction of respiratory tolerance
Respiratory tolerance is a process by which the immune system becomes hyporesponsive to inhaled innocuous environmental antigens. To induce tolerance, we challenged WT and PD-L2[KO] mice intranasally with OVA on days −10, −9, and −8. Mice were then sensitized on day 0 with OVA in alum intraperitoneally and on days 7, 8, and 9 mice were further challenged i.n. with OVA to induce inflammation (Fig. 1C). We first compared the number of T cells responsive to OVA recall after tolerance induction in WT and PD-L2[KO] mice on day 7. Splenocytes were isolated and labeled with CellTrace Violet (CTV) before culture with or without OVA protein for 96 h (Fig. 1D, E). Importantly, we found that T cells from mice that received no antigen in vivo were unresponsive to antigen recall, as were all conditions with no OVA ex vivo. The proliferation of CD3+ CD4+ T cells in response to OVA was however significantly higher in PD-L2[KO] tolerized mice as compared to WT tolerized mice (Fig. 1E). Similarly, we found higher ³[H]-thymidine incorporation in cells from PD-L2[KO] mice and higher levels of IL-2 production, together suggesting a higher proliferation of OVA-responsive T cells—and subsequently a lower tolerance—in the absence of PD-L2 (Supplementary Fig. 2A, B). To further characterize the effects of PD-L2 on airway tolerance, we measured airway hyperreactivity (AHR) in response to increasing doses of methacholine following challenge of mice on days 7, 8, and 9 with OVA i.n. (Fig. 1F). Compared to control mice that received no antigen (PBS), we first found that WT mice challenged with OVA induced significantly higher lung resistance. However, strikingly, PD-L2[KO] mice challenged with OVA i.n. showed higher lung resistance as compared to WT mice. In line with these findings, the numbers of CD45+ cells found in the BAL were higher in PD-L2[KO] mice when compared to controls, and at baseline in mice that received no antigen (Fig. 1G). We further measured significantly higher numbers of inflammatory cells in the BAL of PD-L2[KO] mice, including eosinophils (Fig. 1H). Interestingly, we made similar observations using an anti-PD-L2 blocking antibody (clone 3.2), suggesting that the PD-L2-mediated control of tolerance is inducible (Supplementary Fig. 2C–H). In these experiments, WT mice were treated i.p. with a PD-L2 blocking antibody or the corresponding isotype control. Similarly, we found that splenocytes from anti-PD-L2-treated mice proliferated more compared to controls (Supplementary Fig. 2C–E), as lung resistance and associated airway inflammation were both higher upon PD-L2 blockade (Supplementary Fig. 2F–H). Together, our findings show that PD-L2 contributes to respiratory tolerance in response to OVA.

### PD-L2 regulates iTreg differentiation
One mechanism by which excessive reactivity of the immune system is controlled is through the induction and activation of Tregs in the periphery[22]. Treg development and function are tightly regulated, as studies have shown that multiple pathways can control Treg development, including in particular PD-1 ligand PD-L1[18]. We therefore next assessed whether PD-L2 could modulate iTreg development and function by using a PD-L2Fc to mimic PD-1/PD-L2 interactions in vitro. We cultured pure populations of CD3+CD4+CD44−CD62L+Foxp3− naive WT T cells in the presence of IL-2, TGF-β, and CD3/CD28 stimulation for 72 h, in the presence of PD-L2Fc or corresponding control (Fig. 2A). Although Foxp3 was efficiently induced in both conditions (Fig. 2B), we found that PD-1/PD-L2 interactions resulted in an increase in iTreg numbers compared to controls following 72 h of culture (Fig. 2C). In line with this finding, PD-1/PD-L2 interactions resulted in an increase of Foxp3 expression (Fig. 2D) as well as IL-10 secretion in the culture supernatants as a measure of iTreg function (Fig. 2E). Compared to Th1 or Th2 cells, Tregs use mitochondrial metabolism and oxidative

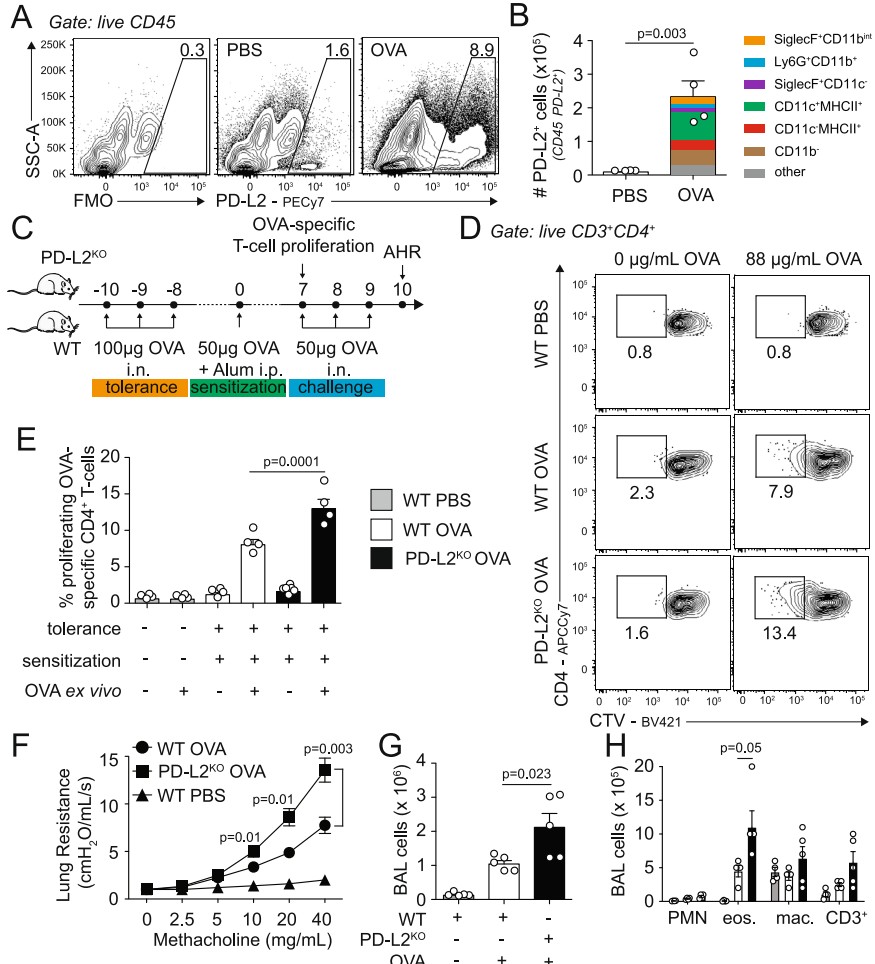

**Fig. 1 | PD-L2 deficiency prevents the induction of respiratory tolerance.**
**A–B** BALB/c (WT) mice were intranasally challenged on 3 consecutive days with 100 μg ovalbumin (OVA) or PBS. On day 4, lungs were collected and the expression of PD-L2 was measured by flow cytometry. **A** Representative flow cytometry plots representing the expression of PD-L2 in CD45+ lung cells **B** Numbers of PD-L2-expressing lung cells presented as mean ± SEM. FMO: "full minus one" staining control. Colors show the frequency of the indicated immune cells within total CD45+PD-L2+ lung cells on day 4 following OVA i.n. challenge. other: mix of all cells not mentioned. *n* = 4 biologically independent mice. **C** BALB/c (WT) and PD-L2KO mice were i.n. challenged on days −10, −9, and −8 with 100 μg ovalbumin (OVA). On day 0, mice were intraperitoneally sensitized with 50 μg OVA emulsified in alum (1:1) and on days 7, 8 and 9 mice were further i.n. challenged with 50 μg OVA i.n. WT mice with no antigen challenge but treated with PBS intranasally were used as controls.

**D** Representative plots of CTV dilution on day 7 within CD3+CD4+ splenocytes following 4 days of culture with or without OVA-peptide. **E** Frequencies of proliferating OVA-specific CD3+CD4+ T cells presented as mean ± SEM. *n* = 4 biologically independent mice. **F** Lung resistance in response to increasing doses of methacholine measured in restrained ventilated mice at day 10. **G** Number of total CD45+ bronchoalveolar lavage (BAL) fluid cells. **H** Numbers of CD11b+ Ly6G+ neutrophils (PMN), CD11c− SiglecF+ eosinophils (eos.), CD11c+ SiglecF+ CD64+ macrophages (mac.) and CD3+ T cells. *n* = 5 biologically independent mice. Data are representative of two independent experiments and are presented as means ± SEM. Source data are provided as a Source Data file. A two-tailed Student's *t* test for unpaired data were applied for comparisons between two groups, except for multi-group comparisons where Tukey's multiple comparison one-way ANOVA tests were used. Mouse image provided with permission from Servier Medical Art.

---

phosphorylation (OXPHOS) to generate energy, together favoring Treg stability and function[23]. Remarkably, we found that although PD-1/PD-L2 interactions did not modulate basal respiration (Fig. 2F, G), it increased spare respiratory capacity in response to FCCP (Fig. 2F, H) and corresponding ATP production (Fig. 2F, I), overall indicating increased mitochondrial activity and oxidative energy production. The transcription factor Foxp3 controls differentiation and function of Tregs[24], as chronic inflammation is associated with reduced Foxp3 expression, function, and loss of phenotypic stability[25]. Furthermore, Tregs possess a distinct DNA methylation pattern that besides the *Foxp3* promoter, is mainly controlled in the three conserved non-coding sequences (CNS) within the locus: CNS1, CNS2, and CNS3 (Fig. 2J). In particular, CNS2 is highly rich in CpG motifs, and demethylation in the Treg-specific demethylated region (TSDR) is correlated with stable expression and stability of Foxp3[26]. Genomic DNA was therefore extracted from iTregs and the degree of methylation of

CpG#19, CpG#20, CpG#21, and CpG#22 within the TSDR region was determined by bisulfite sequencing as described in the methods. We found that PD-1/PD-L2 interactions remarkably decreased DNA methylation of the TSDR region (Fig. 2K), with the frequency of methylation of all but one of the analyzed CpGs modulated by PD-L2Fc treatment (Fig. 2L). Together, our findings, therefore, suggest that PD-L2 binding to PD-1 on Tregs efficiently controls iTreg development and stability in vitro.

**PD-L2 controls Treg homeostasis**
We next sought to determine the function of PD-L2 on Treg homeostasis in vivo using the PD-L2 Foxp3GFP mouse model. Tregs are generated via two distinct developmental programs: the CD25+ Treg cell progenitors (CD25+ TregP cells) and the Foxp3lo Treg cell progenitors (Foxp3lo TregP cells)[27] (Fig. 3A). In the absence of PD-L2, we did not find any differences in neither CD25+ nor Foxp3lo TregP cells, suggesting

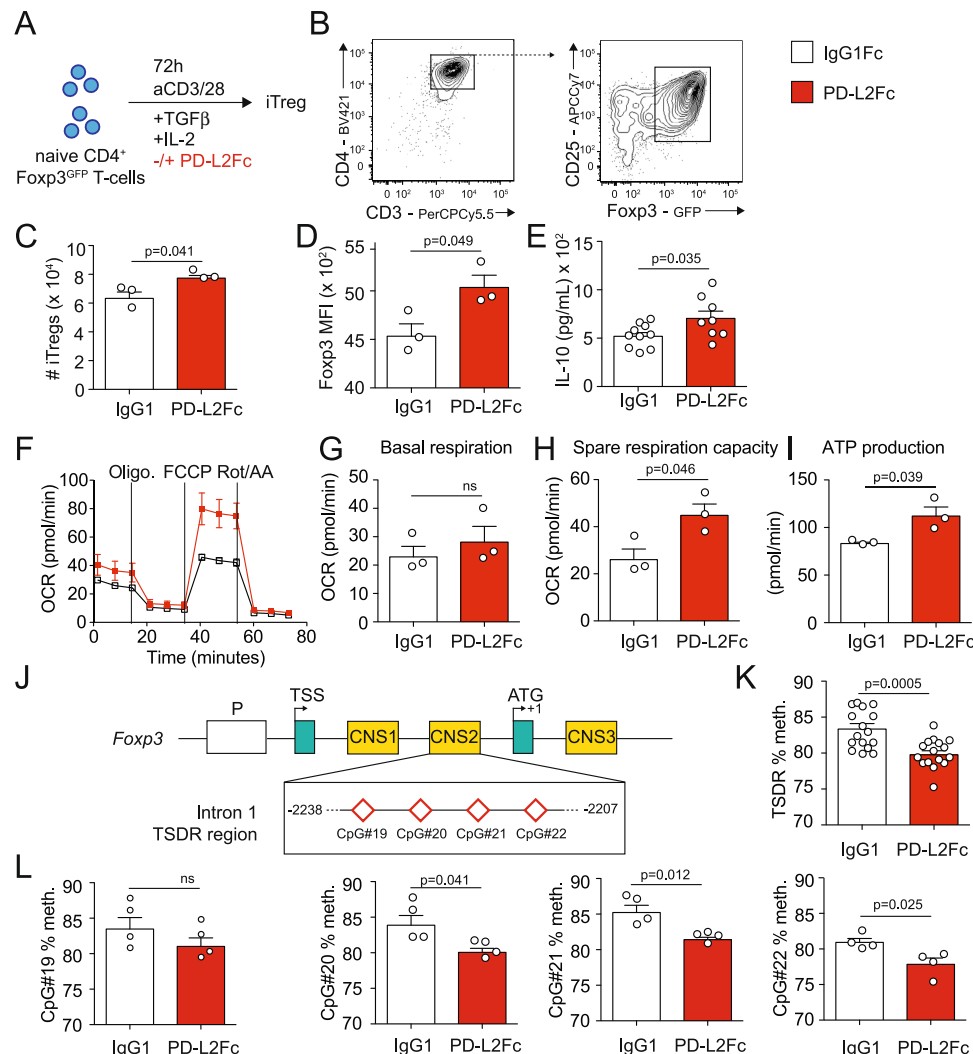

**Fig. 2 | PD-L2 regulates iTreg differentiation. A** Naive CD3+CD4+CD44−CD62L+Foxp3− naive T cells were isolated from the spleens of WT Foxp3GFP and cultured for 72 h in the presence of CD3/CD28 beads, IL-2 and TGF-β. Culture plates were coated with 5 µg/mL PD-L2Fc or control IgG1Fc for 30 min prior culture. **B** Representative flow cytometry plots of Foxp3 induction in CD3+CD4+CD24+Foxp3GFP+ iTregs after 72 h of culture. **C** Absolute numbers of iTregs after 72 h of culture, presented as mean ± SEM. n = 3 biologically independent samples. **D** Foxp3 mean fluorescence intensity (MFI) of iTregs after 72 h of culture, presented as mean ± SEM. n = 3 biologically independent samples. **E** IL-10 secretion by iTregs after 72 h of culture, presented as the mean ± SEM. n = 8 biologically independent samples. **F** Following culture, cells were transferred to PDL-coated HS mini plates and a Mito Stress test was performed to measure oxygen levels in the culture

medium. Oxygen consumption rate (OCR) in response to Oligomycin (oligo.), FCCP and Rotenone (Rot/AA) sequential injections. n = 3 biologically independent samples. **G** Basal respiration in PD-L2Fc or control IgG1Fc groups. **H** Spare respiratory capacity in PD-L2Fc or control IgG1Fc groups. **I** ATP production in PD-L2Fc or control IgG1Fc groups. **J** Schematic of the *Foxp3* promoter region, showing the intron 1 TSDR region containing CpG#19, CpG#20, CpG#21, and CpG#22 within the CNS2. **K** Average methylation of CpG#19, CpG#20, CpG#21, and CpG#22 in TSDR region. **L** Methylation of CpG#19, CpG#20, CpG#21, and CpG#22 in TSDR region. n = 4 biologically independent samples. Data are representative of 2 independent experiments and are presented as means ± SEM. Source data are provided as a Source Data file. A two-tailed Student's *t* test for unpaired data was applied for comparisons between two groups.

that Treg thymic differentiation is not affected by the lack of PD-L2 (Fig. 3B). We further quantified the numbers of splenic CD3+CD4+CD25+Foxp3+ Tregs at homeostasis in PD-L2KO Foxp3GFP mice compared to Foxp3GFP control mice (Fig. 3C). Surprisingly, we found that mice lacking PD-L2 had a significant reduction in total splenic Foxp3+ Tregs at steady state (Fig. 3D). Since we showed that CD11c+ MHCII+ cells in the lungs accounted for the majority of PD-L2 expressing cells, we next addressed whether adoptive transfer of bone marrow-derived dendritic cells (BMDCs) expressing−or not−PD-L2 could restore splenic Foxp3+ Tregs. We, therefore, generated BMDCs from WT mice, which we incubated for 24 h with or without recombinant mouse (rm)IL-4 to induce PD-L2 (Fig. 3E). As described previously, IL-4 strongly induces PD-L2 expression (PD-L2high BMDC) compared to controls (PD-L2low BMDC) (Fig. 3F, G)[16]. We transferred

PD-L2low or PD-L2high BMDCs to both WT Foxp3GFP and PD-L2KO Foxp3GFP recipient mice and measured the numbers of total splenic Foxp3+ Tregs 3 days post transfer (Fig. 3H). In WT recipients, we did not find any significant effects of neither PD-L2low nor PD-L2high BMDCs on the numbers of splenic Tregs compared to untreated mice (Fig. 3I). However remarkably, the adoptive transfer of PD-L2high BMDCs to PD-L2KO Foxp3GFP recipients significantly increased the numbers of total splenic Tregs 3 days after transfer compared to untreated mice, reaching levels comparable to WT control mice. To further understand the mechanisms underlying these differences, we next incubated WT or PD-L2KO BMDCs for 24 h with rmIL-4 and subsequently adoptively transferred either PD-L2high BMDCs or PD-L2null BMDCs, respectively, to PD-L2KO Foxp3GFP recipients (Fig. 3J). Confirming our previous findings, adoptive transfer of PD-L2high BMDCs restored the numbers of total

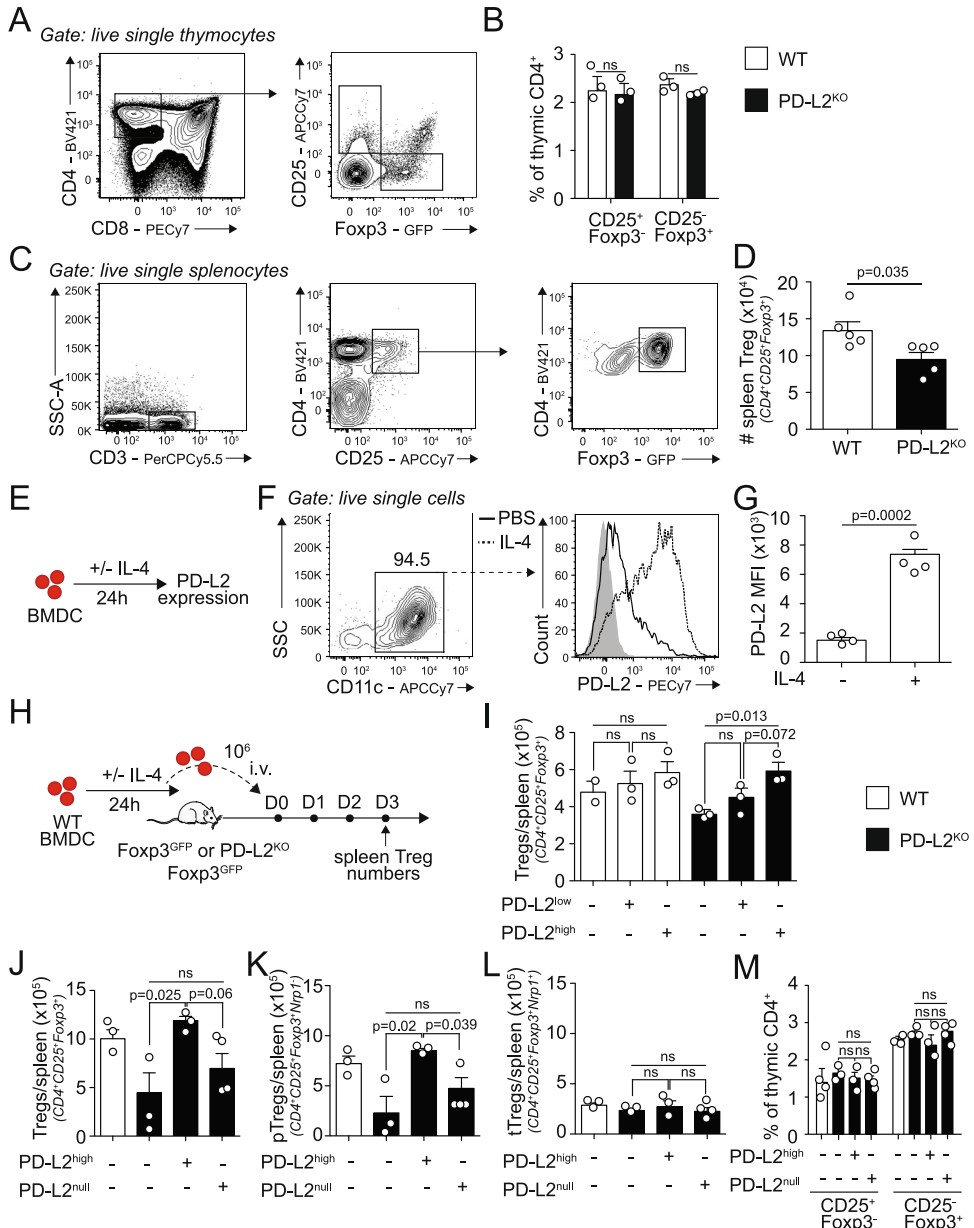

**Fig. 3 | PD-L2 controls Treg homeostasis. A** Thymus of Foxp3$^{GFP}$ and PD-L2$^{KO}$ Foxp3$^{GFP}$ mice were collected and analyzed by flow cytometry. Representative plots of CD25$^+$ (CD25$^+$ TregP cells) and Foxp3$^{lo}$ Treg cell progenitors (Foxp3$^{lo}$ TregP cells) analyzed based on CD25 and Foxp3$^{GFP}$ expression. **B** Corresponding quantitation presented as mean ± SEM. $n = 3$ biologically independent mice. **C** Spleens of Foxp3$^{GFP}$ and PD-L2$^{KO}$ Foxp3$^{GFP}$ mice were collected and analyzed by flow cytometry. Representative plots of CD3$^+$CD4$^+$CD25$^+$Foxp3$^{GFP+}$ total Tregs. **D** Numbers of total Tregs in WT and PD-L2$^{KO}$ mice presented as mean ± SEM. $n = 5$ biologically independent mice. **E** BALB/c (WT) BMDC were differentiated in vitro for 7–10 days, plated and further cultured with or without 50 ng/mL IL-4 for 24 h to measure PD-L2 expression by flow cytometry. **F** Representative plots of CD11c$^+$ BMDC, expression of PD-L2. **G** Numbers of PD-L2-expressing CD11c$^+$ BMDC presented as mean ± SEM. $n = 4$ biologically independent samples. **H** PD-L2$^{low}$ and PD-L2$^{high}$ BMDCs were intravenously injected or not to WT or PD-L2$^{KO}$ Foxp3$^{GFP}$ mice on day 0, and on day 3 the spleens of recipient mice were analyzed for CD3$^+$CD4$^+$CD25$^+$Foxp3$^+$ total Tregs by flow cytometry. **I** Numbers of CD3$^+$CD4$^+$CD25$^+$Foxp3$^{GFP+}$ total Tregs per spleen at day 3 following BMDC transfer. $n = 3$ biologically independent mice. **J**–**M** PD-L2$^{high}$ or PD-L2$^{null}$ BMDCs were i.v. injected or not to WT or PD-L2$^{KO}$ Foxp3$^{GFP}$ mice on day 0, and on day 3 the spleens of recipient mice were analyzed for Tregs by flow cytometry. **J** Numbers of CD3$^+$CD4$^+$CD25$^+$Foxp3$^{GFP+}$ total Tregs. **K** Numbers of CD3$^+$CD4$^+$CD25$^+$Foxp3$^{GFP+}$Nrp1$^-$ pTregs. **L** Numbers of CD3$^+$CD4$^+$CD25$^+$Foxp3$^{GFP}$$^+$Nrp1$^+$ tTregs. **M** Thymus of recipient mice were collected and analyzed by flow cytometry, and the frequencies of CD25$^+$ TregP cells and Foxp3$^{lo}$ TregP cells was analyzed based on CD25 and Foxp3$^{GFP}$ expression, presented as mean ± SEM. $n = 3$ biologically independent mice. Data are representative of three independent experiments and are presented as means ± SEM. Source data are provided as a Source Data file. A two-tailed Student's $t$ test for unpaired data was applied for comparisons between two groups, except for multi-group comparisons where Tukey's multiple comparison one-way ANOVA tests were used. Mouse image provided with permission from Servier Medical Art.

splenic Tregs to numbers comparable to WT controls. However, remarkably, the adoptive transfer of PD-L2$^{null}$ BMDCs failed to restore the numbers of splenic Tregs, suggesting that PD-L2 on BMDCs—and not other dendritic cells characteristics—is required to restore splenic Treg numbers. Although there is currently no reliable marker to

distinguish tTregs to pTregs in vivo, studies have determined that Nrp1—a receptor for TGFβ1—is expressed on thymus-derived tTregs but not peripherally derived pTregs under homeostatic conditions[8]. Although PD-L2$^{null}$ BMDCs failed to modulate splenic pTreg numbers, adoptive transfer of PD-L2$^{high}$ BMDCs remarkably increased the

numbers of splenic pTregs in comparison to controls (Fig. 3K). Of note, the numbers of splenic tTregs (Fig. 3L) as well as thymic Treg progenitors (Fig. 3M) were unaffected, suggesting that PD-L2 controls peripherally derived Tregs. Together, our findings, therefore, suggest that although mice lacking PD-L2 have decreased numbers of splenic Foxp3[+] Tregs at steady state, these numbers are restored by the adoptive transfer of BMDCs expressing high levels of PD-L2. Furthermore, this effect was PD-L2-specific, and remarkably only affected induction of Tregs in the periphery.

### PD-L2 regulates pTreg numbers, function, and Foxp3 stability

We next sought to characterize the effects of PD-L2 on splenic tTreg and pTreg numbers and function under homeostatic conditions. Although we found similar numbers of splenic Foxp3[+]Nrp1[+] tTregs in WT and PD-L2[KO] mice, there notably was a significant reduction in Foxp3[+]Nrp1[−] pTregs numbers in the spleen of mice constitutively lacking PD-L2 (Fig. 4A, B). On average, ~60% of Tregs in the spleen were pTregs in WT mice, compared to ~40% in the absence of PD-L2 (Supplementary Fig. 3A, B). Remarkably, this tendency was also observed in both peripheral lymph nodes as well as the lungs (Supplementary Fig. 3C), suggesting that PD-L2 is required for the induction of Tregs from the periphery. Importantly, we obtained similar results when representing total splenic Tregs, pTregs, and tTregs as a frequency of CD3[+] cells (Supplementary Fig. 3D, E). We found that pTregs isolated from PD-L2[KO] mice produced significantly less IL-10 compared to controls upon activation (Fig. 4C). Although tTregs produced less IL-10 compared to pTregs, we further observed a tendency of lower IL-10 secretion in tTregs isolated from PD-L2[KO] mice compared to controls (Fig. 4D). Interestingly, both pTregs and tTregs isolated from PD-L2[KO] mice produced less latency-associated peptide (LAP) as a measure of TGF-β (Supplementary Fig. 3F). We next assessed pTreg function by measuring their capacity to suppress effector T-cell proliferation in culture. We therefore cultured CD3[+]CD4[+]CD44[−]CD62L[+] naive T cells with pTregs for 72 h and measured T cell proliferation by quantifying thymidine incorporation from harvested cells (Fig. 4E). As controls, effector T cells alone (0:1) proliferated efficiently, and addition of WT pTregs at increasing ratios gradually suppressed T cell proliferation, as evidenced by the decreased thymidine incorporation after culture (Fig. 4F). However, strikingly, we found that pTregs isolated from PD-L2[KO] Foxp3[GFP] significantly suppressed less effector T cell division compared to WT pTreg controls (Fig. 4F). To understand the mechanisms of decreased function and suppressive capacity of pTregs isolated from PD-L2[KO] mice, we next evaluated Foxp3 expression and DNA methylation status of the Foxp3 TSDR region of pTregs isolated from Foxp3[GFP] and PD-L2[KO] Foxp3[GFP] mice (Fig. 4G). Importantly, we first found that the expression of Foxp3 within pTregs was remarkably decreased in the absence of PD-L2 (Fig. 4H). On average, the TSDR region of pTregs isolated from PD-L2[KO] Foxp3[GFP] pTregs was significantly more methylated compared to controls (Fig. 4I). In particular, the methylation status of each of the four studied CpGs was increased in pTregs isolated from PD-L2[KO] Foxp3[GFP] compared to controls (Fig. 4J). To address whether PD-L2 was involved in Treg Foxp3 stability in vivo, we adoptively transferred WT Foxp3[GFP] iTreg to WT BALB/c or PD-L2[KO] hosts intravenously and measured GFP signals in splenocytes 3 days after transfer (Fig. 4K). Remarkably, we found significantly less Foxp3[GFP] Tregs in PD-L2[KO] mice compared to WT control hosts, suggesting that Treg stability is affected in the absence of PD-1/PD-L2 interactions in the recipient mice (Fig. 4L, M). Together, our findings therefore suggest that PD-L2 is required for pTreg development in the periphery, favoring Foxp3 stability and function.

### PD-L2 modulates pTreg signature genes and mitochondrial function

Cellular metabolism is closely related to Treg cell stability, as studies have shown that the expression of Foxp3 reprograms T-cell

metabolism by suppressing glycolysis[28]. We, therefore, focused next on the role of PD-L2 on major Treg metabolic sources fueling the tricarboxylic acid (TCA) cycle and related energy production[29]. pTreg cells were FACS-sorted from the spleens of Foxp3[GFP] and PD-L2[KO] Foxp3[GFP] mice and incubated for 24 h with CD3/CD28 beads, before mRNA was isolated and RNA-sequencing (RNA-seq) was performed (Fig. 5A). We found a significant decrease in genes associated with Treg function in pTregs isolated from PD-L2[KO] Foxp3[GFP] mice compared to controls (Fig. 5B). We selected 33 genes based on a previous publication[30] that represent well-known Treg signature markers and in particular found that *il10* was downregulated in pTregs isolated from PD-L2[KO] Foxp3[GFP] mice compared to controls, as well as the immune checkpoint receptor *Ctla4*. Importantly, we found that *Pdcd1*—encoding for PD-L2 receptor PD-1—was highly but not differentially expressed in WT and pTregs isolated from PD-L2[KO] mice. Furthermore, although PD-L2 can also bind to novel receptor RGMb, we did not find detectable levels of *Rgmb* transcripts in either WT or pTregs isolated from PD-L2[KO] mice[31]. We next found that genes included in the fatty acid oxidation (FAO) (Fig. 5C), amino acid (AA) degradation (Fig. 5D), and the TCA cycle (Fig. 5E) pathways were all downregulated in PD-L2[KO] Foxp3[GFP] pTregs at the transcriptomic level. In particular carnitine palmitoyl transferase 1a (*Cpt1a*)—the enzyme allowing entry of acyl groups into the mitochondria during FAO—was downregulated (Fig. 5C). Surprisingly, however, we did not find significant modulations in the glycolysis (Fig. 5F) or Pentose Phosphate Pathway (PPP) pathways (Fig. 5G). We therefore next assessed cellular bioenergetics by measuring cellular respiration of pTregs isolated from Foxp3[GFP] and PD-L2[KO] Foxp3[GFP] upon CD3/CD28 stimulation (Fig. 5H–K). We found that pTregs isolated from PD-L2[KO] Foxp3[GFP] showed decreased basal respiration (Fig. 5H, I), spare respiratory capacity in response to FCCP (Fig. 5H, J), and corresponding respiration-coupled ATP production (Fig. 5H, K), overall indicating decreased mitochondrial activity and oxidative energy production. To confirm a defect in mitochondrial function, we next quantified mitochondrial morphology in WT pTregs and pTregs isolated from PD-L2[KO] mice using the MitoTracker Deep Red (MitoTracker[DR]) probe by flow cytometry (Fig. 5L, M). The results show a decrease in MitoTracker[DR] expression in pTregs from mice lacking PD-L2, suggesting that the lack of PD-L2 is associated with a decreased mitochondrial mass in pTregs compared to controls. Overall, our results suggest that pTregs from PD-L2[KO] mice have an impaired TCA cycle and mitochondrial respiration, leading to lower ATP production and causing altered effector functions. We, therefore, tested whether pyruvate, an energy metabolite substituting FA-derived acetyl-coA could directly fuel OXPHOS and restore pTreg-derived IL-10 production (Fig. 5N). Remarkably, we found that pyruvate treatment of pTregs from PD-L2[KO] mice partially restored IL-10 cytokine secretion (Fig. 5O). Together, our transcriptomic and bioenergetic results support the notion that PD-L2 maintains mitochondrial activity and pTreg function. The results further show that administration of pyruvate remarkably partially restores IL-10 production in pTregs from mice lacking PD-L2.

### PD-L2 controls pTreg numbers and function and contributes to respiratory tolerance

We next sought to determine if PD-L2 controlled respiratory tolerance by modulating Treg numbers and function in the lungs. To induce tolerance, we challenged Foxp3[GFP] and PD-L2[KO] Foxp3[GFP] mice i.n. with OVA on days −10, −9, and −8. Mice were then sensitized on day 0 with OVA in alum intraperitoneally and on day 7 we measured Treg numbers and frequencies of pTregs in the lungs (Fig. 6A and Supplementary Fig. 4A). Based on Nrp1 expression, we found that the pTreg frequencies within total Tregs decreased from 68.5% in WT mice to 54.2% in PD-L2[KO] mice (Fig. 6B). The absolute numbers of Foxp3[+] Nrp1[−] pTregs were significantly lower in PD-L2[KO] mice compared to controls (Fig. 6C), although PD-1 was highly expressed in pTregs from both WT

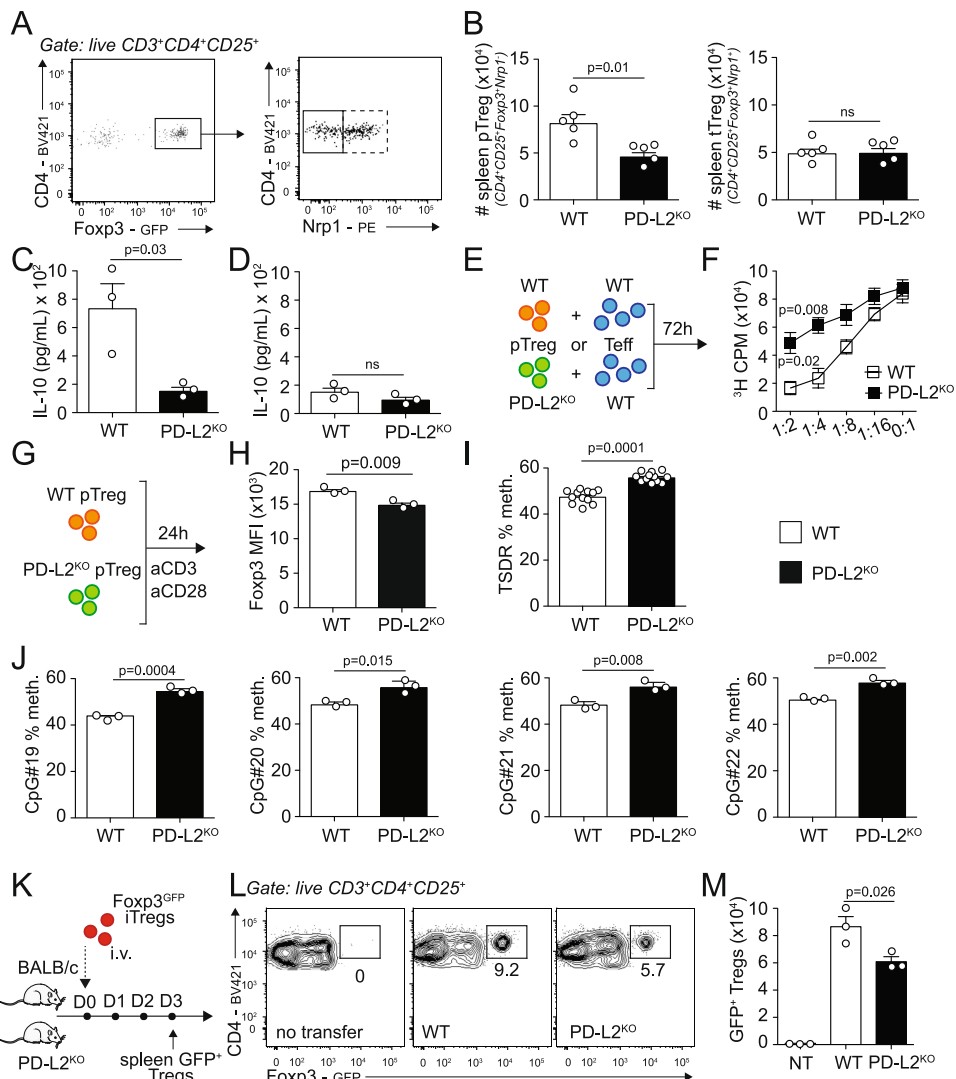

**Fig. 4 | PD-L2 regulates pTreg numbers, function, and Foxp3 stability. A** Spleens of Foxp3$^{GFP}$ and PD-L2$^{KO}$ Foxp3$^{GFP}$ mice were collected and analyzed by flow cytometry. Representative plots of CD3$^+$CD4$^+$CD25$^+$Foxp3$^{GFP+}$Nrp1$^-$ pTregs and CD3$^+$CD4$^+$CD25$^+$Foxp3$^{GFP+}$Nrp1$^+$ tTregs. **B** Quantitation of pTreg and tTreg numbers in the spleen presented as means ± SEM. $n = 5$ biologically independent mice. **C–D** pTregs and tTregs–gated as in (**A**)–were FACS-sorted from the spleens of Foxp3$^{GFP}$ and PD-L2$^{KO}$ Foxp3$^{GFP}$ mice and cultured for 24 h with CD3/CD28 beads, IL-2 and TGF-β. IL-10 was measured by ELISA in the culture supernatants. **C** Levels of IL-10 in pTreg cultures. $n = 3$ biologically independent samples. **D** Levels of IL-10 in tTreg cultures. **E** CD3$^+$CD4$^+$CD44$^-$CD62L$^+$ naive T cells were isolated from the spleens of BALB/c mice and co-cultured with FACS-sorted CD3$^+$CD4$^+$CD25$^+$Foxp3$^{GFP}$$^+$Nrp1$^-$ splenic pTregs isolated from Foxp3$^{GFP}$ or PD-L2$^{KO}$ Foxp3$^{GFP}$ mice for 72 h. Cell proliferation was assessed by measuring thymidine incorporation. **F** $^3$H thymidine incorporation after 72 h of culture. $n = 4$ biologically independent samples. **G** CD3$^+$CD4$^+$CD25$^+$Foxp3$^{GFP+}$Nrp1$^-$ splenic pTregs cells were FACS-sorted from the spleens of Foxp3$^{GFP}$ and PD-L2$^{KO}$ Foxp3$^{GFP}$ mice and incubated for 24 h with CD3/CD28 beads, IL-2 and TGF-β. Genomic DNA was extracted, followed by bisulfite

conversion, purification, cloning, and the degree of methylation of CpG#19, CpG#20, CpG#21, and CpG#22 within the TSDR region was determined by bisulfite sequencing. **H** Foxp3$^{GFP}$ expression within WT and PD-L2$^{KO}$ pTregs. $n = 3$ biologically independent samples. **I** Average methylation of CpG#19, CpG#20, CpG#21, and CpG#22 in TSDR region. $n = 3$ biologically independent samples. **J** Methylation of CpG#19, CpG#20, CpG#21, and CpG#22 in TSDR region. **K** 10$^6$ WT Foxp3$^{GFP}$ iTregs were adoptively transferred to WT BALB/c or PD-L2$^{KO}$ hosts on day 0 and on day 3, the frequencies of GFP$^+$ Tregs were analyzed in the spleen of recipients. **L** Representative plots of CD3$^+$CD4$^+$CD25$^+$Foxp3$^{GFP+}$ Tregs. **M** Numbers of transferred Tregs in the spleen presented as mean ± SEM. $n = 3$ biologically independent mice. Data are representative of two independent experiments and are presented as means ± SEM. Source data are provided as a Source Data file. A two-tailed Student's $t$ test for unpaired data was applied for comparisons between two groups, except for multi-group comparisons where Tukey's multiple comparison one-way ANOVA tests were used. Mouse image provided with permission from Servier Medical Art.

and PD-L2$^{KO}$ (Fig. 6D). Importantly, we further found that the expression of Foxp3 was significantly reduced in pTregs from PD-L2$^{KO}$ mice, suggesting that Foxp3 stability is affected (Fig. 6E). Of note, the numbers of Foxp3$^+$ Nrp1$^+$ tTregs were not significantly reduced in the lungs of PD-L2$^{KO}$ mice (Supplementary Fig. 4B). We next generated an in vitro model to further assess the role of PD-L2 in peripherally induced Treg function and metabolism in the context of inflammation. WT and PD-L2$^{KO}$ BMDC were pulsed with OVA and cultured with CD3$^+$CD4$^+$CD44$^-$CD62L$^+$KJ1-26$^+$ naive T cells in the presence of IL-2,

TGF-β, and CD3/CD28 stimulation to induce Foxp3 for 72 h (Fig. 6F). Although Foxp3 was efficiently induced in T cells (Fig. 6G), we found that T cells cultured with PD-L2$^{KO}$ BMDC expressed significantly less Foxp3 compared to controls (Fig. 6H). Furthermore, we observed a decreased secretion of IL-10 in the culture supernatant (Fig. 6I), together confirming that PD-L2 controls Foxp3 expression and iTreg function. Finally, we found that T cells cultured with PD-L2$^{KO}$ BMDC accumulated more BODIPY$^{493/503}$ neutral lipids, confirming a defect in FAO in iTregs in the absence of PD-L2 (Fig. 6J). Since we previously

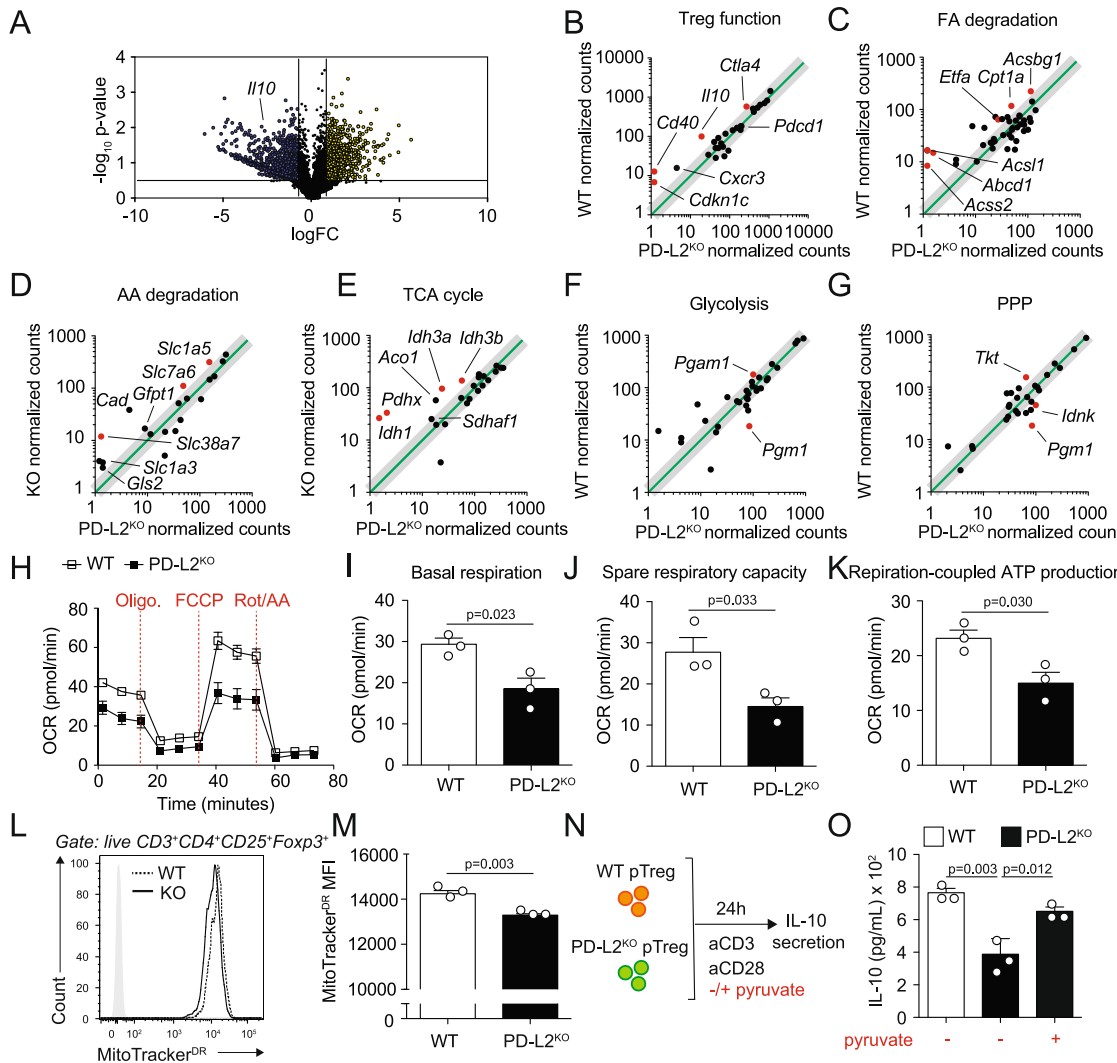

**Fig. 5 | PD-L2 modulates pTreg signature genes and metabolic function.**
**A** CD3⁺CD4⁺CD25⁺Foxp3^GFP⁺Nrp1⁻ splenic pTregs cells were FACS-sorted from the spleens of Foxp3^GFP and PD-L2^KO Foxp3^GFP mice and incubated for 24 h with CD3/CD28 beads, IL-2 and TGF-β. Cells were then lysed and mRNA was isolated to perform RNA-sequencing (RNA-seq). Volcano plot representation of differentially regulated genes in WT and PD-L2^KO pTregs. Dotted lines represent 2FC (x axis) and $p < 0.05$ (y axis) cutoffs. **B–G** Differentially regulated genes in selected pathways. **B** Treg function pathway. **C** Fatty acid (FA) degradation pathway. **D** Amino acid (AA) degradation pathway. **E** TCA cycle pathway. **F** Glycolysis pathway. **G** Pentose Phosphate Pathway (PPP) pathway. All red dots represent genes whose differences are $p < 0.05$. **H–K** CD3⁺CD4⁺CD25⁺Foxp3^GFP⁺Nrp1⁻ splenic pTreg cells were FACS-sorted from the spleens of Foxp3^GFP and PD-L2^KO Foxp3^GFP mice and incubated for 24 h with CD3/CD28 beads, IL-2 and TGF-β. Following culture, cells were transferred to PDL-coated HS mini plates and a Mito Stress test was performed to measure oxygen levels in the culture medium. **H** Oxygen consumption rate (OCR) in response to Oligomycin (oligo.), FCCP, and Rotenone (Rot/AA) sequential

injections. **I** Basal respiration in WT and PD-L2^KO pTregs. $n = 3$ biologically independent samples. **J** Spare respiratory capacity in WT and PD-L2^KO pTregs. **K** Respiration-coupled ATP production. **L** Representative flow cytometry plots of MitoTracker^DR expression within WT and PD-L2^KO pTregs after 24 h of culture. **M** Quantitation of MitoTracker^DR expression within WT and PD-L2^KO pTregs after 24 h of culture, presented as mean ± SEM. $n = 3$ biologically independent samples. **N** CD3⁺CD4⁺CD25⁺Foxp3^GFP⁺Nrp1⁻ splenic pTregs cells were FACS-sorted from the spleens of Foxp3^GFP and PD-L2^KO Foxp3^GFP mice and incubated for 24 h with CD3/CD28 beads, IL-2 and TGF-β with or without 2 mM methyl pyruvate. **O** Levels of IL-10 production in the culture supernatants. $n = 3$ biologically independent samples. **H–O** Data are representative of two independent experiments and are presented as means ± SEM. Source data are provided as a Source Data file. A two-tailed Student's $t$ test for unpaired data was applied for comparisons between two groups, except for multi-group comparisons where Tukey's multiple comparison one-way ANOVA tests were used.

observed that pyruvate partially restored IL-10 production in Tregs, we next assessed whether pyruvate could restore respiratory tolerance (Fig. 6K–M). Remarkably, we found that compared to untreated PD-L2^KO mice, treatment with pyruvate decreased lung resistance in response to methacholine (Fig. 6L) as well as inflammatory cells such as eosinophils (Fig. 6M) to levels comparable to the WT controls. Such findings, therefore, suggest that pyruvate treatment is able to restore respiratory tolerance in PD-L2^KO mice in vivo. Together, our findings, therefore, support the notion that PD-L2 contributes to respiratory tolerance by maintaining oxidative metabolism, Foxp3 stability, and suppressive function in peripherally induced Tregs.

## Discussion

Over the years, PD-1 and its ligands PD-L1 and PD-L2 have been shown to be implicated in many diseases including allergic diseases and asthma[32]. Both PD-L1 and PD-L2 regulate airway inflammation and AHR. Notably, activation of the PD-1/PD-L2 pathway in the lungs protects against the initiation and progression of airway inflammation[16]. Furthermore, Foxp3⁺ Tregs highly express PD-1 and ligand PD-L1, as this pathway is crucial for Treg generation and tolerance[14]. In particular, iTreg cell differentiation, maintenance, and function are induced by PD-L1, sustaining and increasing the expression of Foxp3, together tipping the balance towards immunologic tolerance[18]. However, the

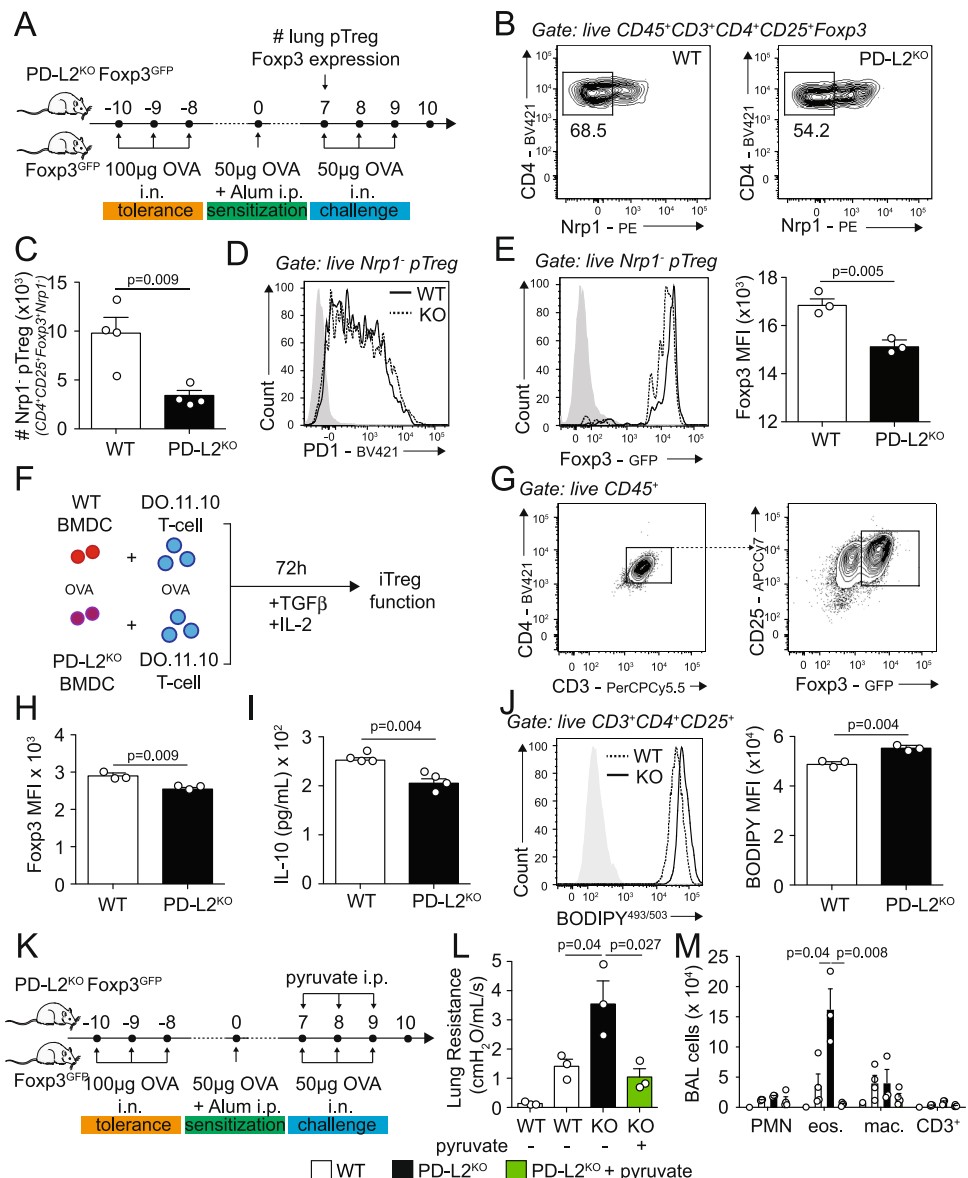

**Fig. 6 | PD-L2 controls pTreg numbers and function to induce respiratory tolerance. A** Foxp3$^{GFP}$ and PD-L2$^{KO}$ Foxp3$^{GFP}$ mice were i.n. challenged on days −10, −9 and −8 with 100 μg ovalbumin (OVA). On day 0, mice were intraperitoneally sensitized with 50 μg OVA emulsified in alum (1:1) and CD3$^+$CD4$^+$CD25$^+$Foxp3$^{GFP+}$Nrp1$^-$ pTreg numbers were analyzed by flow cytometry on day 7. **B** Representative plots of pTregs in the lungs of Foxp3$^{GFP}$ and PD-L2$^{KO}$ Foxp3$^{GFP}$ mice. **C** Numbers of pTregs in the lungs of Foxp3$^{GFP}$ and PD-L2$^{KO}$ Foxp3$^{GFP}$ mice presented as mean ± SEM. $n = 4$ biologically independent mice. **D** Representative plots of PD-1 expression in lung pTregs from Foxp3$^{GFP}$ and PD-L2$^{KO}$ Foxp3$^{GFP}$. **E** Representative plots of Foxp3 expression in lung pTregs from Foxp3$^{GFP}$ and PD-L2$^{KO}$ Foxp3$^{GFP}$ mice and corresponding quantitation presented as mean ± SEM. $n = 3$ biologically independent mice. **F** BALB/c (WT) and PD-L2$^{KO}$ BMDC were differentiated in vitro for 7–10 days and cultured with FACS-sorted CD4$^+$CD44$^-$CD62L$^+$KJ1-26$^+$ naive T cells in the presence of IL-2, TGF-β, 100 ng/mL OVA peptide and CD3/CD28 stimulation to induce Treg differentiation for 72 h at 1:20 ratio. **G** Representative plot of Foxp3 expression. **H** Foxp3 expression in T cells incubated with either WT or PD-L2$^{KO}$ BMDC,

presented as mean ± SEM. $n = 3$ biologically independent mice. **I** Levels of IL-10 in the supernatants, presented as mean ± SEM. $n = 4$ biologically independent samples. **J** Representative plots of BODIPY$^{493/503}$ expression in T cells incubated with either WT or PD-L2$^{KO}$ BMDC and corresponding quantitation presented as mean ± SEM. $n = 3$ biologically independent samples. **K** Foxp3$^{GFP}$ and PD-L2$^{KO}$ Foxp3$^{GFP}$ mice were challenged as in **A**. On days 7, 8, and 9 mice were further challenged or not with 100 mg/kg ethyl pyruvate intraperitoneally. **L** Lung resistance in response to 40 mg/mL of methacholine. **M** Numbers of CD11b$^+$ Ly6G$^+$ neutrophils (PMN), CD11c$^-$ SiglecF$^+$ eosinophils (eos.), CD11c$^+$ SiglecF$^+$ CD64$^+$ macrophages (mac.) and CD3$^+$ T cells in the BAL fluid. $n = 3$ biologically independent mice. Data are representative of two independent experiments, presented as means ± SEM. Source data are provided as a Source Data file. A two-tailed Student's $t$ test for unpaired data was applied for comparisons between two groups, except for multi-group comparisons where Tukey's multiple comparison one-way ANOVA tests were used. Mouse image provided with permission from Servier Medical Art.

role of the PD-1/PD-L2 pathway in Treg function is less clear. Whereas PD-L1 is highly expressed and induced in CD4$^+$, CD8$^+$, and Foxp3$^+$ Tregs, PD-L2 expression is absent from CD3$^+$ T cells and limited to certain types of antigen-presenting cells upon specific stimulation[14]. For instance, whereas macrophages and dendritic cells do not express PD-L2 under homeostatic conditions, IL-4 stimulation strongly induces

PD-L2 but not PD-L1 expression, suggesting a context-dependent role of PD-L2[15,16]. In the context of asthma, we have previously described the important role of PD-L1 and PD-L2 on lung DCs in modulating asthma pathogenesis[16]. In particular, PD-L2$^{KO}$ mice challenged with OVA intranasally showed significantly higher AHR and airway inflammation compared to controls and PD-L1$^{KO}$ mice, strongly suggesting that PD-

L2 is involved in controlling the magnitude of lung inflammation in such context.

Here, we demonstrate that the PD-1/PD-L2 axis contributes to the development of respiratory tolerance in the lungs. We show that OVA-induced lung inflammation stimulates myeloid cells to express PD-L2 in the lungs. Using a combination of genetically modified mice lacking PD-L2 and antibody blockade, we demonstrate that this expression of PD-L2 is required for tolerance to ovalbumin. Tolerized PD-L2[KO] mice showed increased AHR and lung inflammation when re-challenged with OVA as compared to controls, associated with increased BAL inflammatory cell recruitment. PD-L2[KO] mice further displayed increased CD4[+] T-cell proliferation upon antigen recall ex vivo, suggesting that they are less tolerant compared to controls. To evaluate the contribution of PD-L2 in respiratory tolerance, we assessed the numbers and function of Foxp3[+] Tregs, as studies suggest that Tregs serve as the most critical cellular mechanism for promoting tolerance to antigens inhaled into the airways[33]. Following OVA treatment, we found significantly less Tregs in the lungs of PD-L2[KO] mice compared to controls. Compared to WT BMDCs, PD-L2[KO] BMDCs pulsed with OVA-induced lower Foxp3 expression and IL-10 secretion in iTregs in vitro, suggesting that PD-L2 is required for iTreg development and function. In line with this, we further used a PD-L2Fc to directly assess PD-1/PD-L2 interactions in vitro and found that PD-L2 directly increased the numbers and stability of iTregs, confirming our initial finding that exogenous PD-L2 modulates iTreg cell differentiation and maintenance.

Under homeostatic conditions, there were significantly lower numbers of Tregs in PD-L2[KO] mice compared to controls in the spleen. In particular, the proportion and numbers of Foxp3[+]Nrp1[−] pTregs—but not Foxp3[+]Nrp1[+] tTregs—were affected by the absence of PD-L2. Remarkably, however, the numbers of Tregs in the spleen were restored following adoptive transfer of PD-L2[high] BMDCs, whereas PD-L2[KO] BMDCs failed to restore Treg numbers. We further found that only pTreg numbers were restored by the adoptive transfer of PD-L2[high] BMDCs, as both splenic tTreg numbers and thymic Treg progenitors were unaffected by the absence of PD-L2. pTregs—unlike tTregs—isolated from PD-L2[KO] mice produced less inhibitory cytokines IL-10 and TGF-β upon activation, as they failed to suppress effector T cells as efficiently as WT pTregs ex vivo. It is however noteworthy to point out that since the Nrp1 antibody used to FACS sort pTregs and tTregs is a TGF-β receptor, one limitation of our study is the fact that we cannot exclude TGF-β stimulation effects that may occur in both pTregs and tTregs. Our results are nevertheless consistent with previous findings showing that PD-L2[KO] mice challenged with chicken OVA exhibited increased activation of CD4[+] and CD8[+] T cells compared to controls[34]. Although conversion and maintenance of pTreg function in vivo was shown to depend on PD-L1[35], our results suggest that PD-L2 is also involved in the process. These effects may be explained at least in part by the preferential PD-L2 upregulation by IL-4, as the OVA model of airway inflammation is driven by high levels of IL-4[16]. In line with this, several reports suggest the essential role of IL-4 in supporting Treg-mediated immune suppression[36].

Besides the development of Foxp3[+] Tregs, the process of respiratory tolerance involves multiple mechanisms that include inhibition of the initial antigen-specific T cell expansion, and induction of T-cell anergy, as these processes may occur simultaneously. In line with this, a previous report describes a role for the PD-L2/RGMb pathway in limiting the development of OVA-responsive T cells that normally occurs after respiratory administration of OVA[31]. RGMb is a glycosylphosphatidylinositol-anchored membrane protein shown to be a second binding partner for PD-L2. Although appreciable levels of *RGMb* mRNA can be found in the lungs, its expression is mainly restricted to interstitial macrophages (IM) and alveolar epithelial cells (AEC), both involved in the induction of respiratory tolerance by producing high levels of IL-10 and the modulation of DC activation,

respectively[37,38]. The PD-L2 interaction with RGMb in the lungs may therefore directly or indirectly modulate IM and/or AEC function, thereby inhibiting the maturation of DCs and induction of respiratory tolerance. Importantly, we did not detect *RGMb* mRNA in either WT or PD-L2[KO] Foxp3[+] splenic pTregs upon activation. We, however, found appreciable levels of *Pdcd1*—encoding for PD-1—in both WT and PD-L2[KO] Foxp3[+] splenic pTregs, as pulmonary pTregs in the OVA-tolerance model expressed high levels of PD-1 on the surface. The altered phenotype of pTregs described in our studies is therefore due to the blocking of PD-L2 and PD-1 engagement. Together, our data and previous findings support a role for PD-L2 in the development of airway tolerance to allergens.

Maintenance of Foxp3 expression is essential for Treg function, as Treg stability—the ability to maintain Foxp3 expression and resist acquiring pro-inflammatory markers—is crucial for Treg function[39]. The mechanisms preventing the loss of Foxp3 expression are driven by transcriptional, epigenetic, and post-translational modifications. In particular, demethylation of the TSDR region within the Foxp3 CNS2 element allows for the recruitment of transcription factors that stabilize Foxp3 expression[40]. Although CNS1 and CNS3 elements are involved in thymus and peripheral Treg development, CNS2 is central to maintain Foxp3 expression within Tregs, as CNS2-deficient Tregs were shown to progressively lose Foxp3 expression after transfer into lymphogenic mice[41]. Our results show significantly higher methylation of the TSDR region in pTregs from PD-L2[KO] mice compared to controls in vivo, associated with lower Foxp3 expression. In line with this, adoptively transferred WT Tregs were less stable in PD-L2[KO] compared to WT recipients, as PD-L2Fc treatment reduced methylation of the TSDR and increased iTreg function, suggesting that exogenous PD-L2 signaling is directly impacting Foxp3 stability of Tregs. Reports have described the generation of plastic Th-like Tregs that acquire the expression of transcription factors associated with effector T-cell program[42]. Although the mechanistic connection between Treg stability and plasticity remains incomplete in disease settings, further studies are required to understand the function of the PD-1/PD-L2 axis in the generation of "plastic" Th-like Treg cells.

Cellular metabolism is closely related to Treg cell stability, as compared to other T cells, Tregs use mitochondrial metabolism and OXPHOS for energy production[23,43]. Our results strongly support a decrease in mitochondrial metabolic function in pTregs isolated from PD-L2[KO] mice. We observed a decrease in major signature genes involved in pathways such as FAO, amino-acid degradation, and TCA cycle in pTregs from PD-L2[KO] mice, as our bioenergetic studies confirmed a defect in mitochondrial respiration and ATP production in pTregs isolated from PD-L2[KO] mice. In confirmation of our findings, PD-L2Fc treatment directly induced OXPHOS in iTregs, confirming that PD-L2/PD-1 interactions stimulate this pathway in iTregs. Furthermore, we found a lower mitochondrial mass in pTregs isolated from PD-L2[KO] mice, as evidenced by our MitoTracker assay. Surprisingly, however, we did not see a higher genetic signature in glycolysis or Pentose Phosphate Pathway, suggesting that PD-L2[KO] Tregs do not switch their metabolic program and PD-1/PD-L2 interactions rather favor induction of FAO. Several lines of evidence support the notion that fatty acids can modulate Foxp3 stability. For instance, deletion of the mitochondrial transcription factor A (Tfam) destabilizes Foxp3 expression and enhances TSDR methylation[44]. Furthermore, short-chain fatty acids can enhance Foxp3 expression by inhibiting histone deacetylases known to affect Foxp3 stability[45], whereas acetyl-CoA levels can also directly enhance Foxp3 expression through post-translational modifications[46]. Importantly, we were able to partially restore IL-10 production by treating cells with a stable form of pyruvate ex vivo, which was also remarkably able to restore immune tolerance in vivo. We have recently reported that exogenous pyruvate treatment similarly restores ILC2 effector function[47]. We hypothesize that fatty acids are used during Treg activation as energy substrates for FAO and

OXPHOS and we used pyruvate as an energy metabolite that can substitute FA-derived acetyl-CoA to directly fuel OXPHOS and promote mitochondrial respiration. In confirmation of our findings, various reports show that administration of FAs such as oleic acid or downstream mediator pyruvate is able to stimulate/restore Treg suppressive functions[48,49]. Although further studies are required to delineate the effects of FAO metabolism on Foxp3 stability in the context of PD-L2 regulation, our study for the first time suggests that PD-L2 may affect Foxp3 stability in Treg cells via modulation of cellular metabolism.

Collectively, our study highlights the important role of the PD-1/PD-L2 axis in airway immune tolerance. PD-L2 maintains metabolic activity and Foxp3 stability, together promoting the maintenance of immunoregulatory Foxp3+ pTregs. Importantly, we were able to show that adoptive transfer of PD-L2-expressing BMDCs in PD-L2^KO mice restored Treg − in particular pTregs−numbers, as induction of OXPHOS via treatment with a stable form of pyruvate further restored IL-10 production, Treg suppressive functions and immune tolerance. Targeting the PD-1 pathway has enormous therapeutic potential in diseases including asthma, as agonistic or antagonistic treatments may offer better long-term effects to patients than the current therapies. In particular, we have recently tested a human PD-1 agonist that was able to ameliorate AHR in response to allergen in humanized mice[50]. In the context of allergy therapy, our findings open up avenues to further improve mucosal immunotherapy by including metabolic or costimulatory modulators to effectively modify pTreg suppressive function and establish immune tolerance.

## Methods

All experimentation protocols were approved by the USC Institutional Animal Care and Use Committee (IACUC) and conducted in accordance with the USC Department of Animal Resources' guidelines and the principles of the Declaration of Helsinki.

### Mice

Five-to-eight-week-old female mice were used in this study. Wild type (WT) BALB/cByJ (stock #001026) and DO11.10 mice on a BALB/c background (C.Cg-Tg(DO11.10)10Dlo/J, stock #003303) were purchased from the Jackson Laboratories (Ann Harbor, ME). Foxp3 Green Florescent Protein (Foxp3^GFP)[51] and PD-L2^KO mice[52]−both on a BALB/c background−have been previously reported and were crossed to obtain PD-L2^KO Foxp3^GFP mice. Crossed mice were genotyped following protocols detailed in the original publications describing Foxp3^GFP [51] and PD-L2^KO [52] mice. All mice were bred separately in specific-pathogen-free conditions in the mouse facility at the Keck School of Medicine, University of Southern California (USC) and maintained at a macroenvironmental temperature of 21–22 °C, humidity (48–52%), in a conventional 12:12 light/dark cycle with lights on at 6:00 a.m. and off at 6:00 p.m. When indicated, mice were euthanized by $CO_2$ inhalation, followed by cervical dislocation.

### Respiratory tolerance and antibody treatment

WT/PD-L2^KO or Foxp3^GFP/PD-L2^KO Foxp3^GFP mice received 100 μg LPS-free OVA (Worthington) or PBS intranasally (i.n) on days −10, −9, and −8 to induce tolerance. On day 0, mice were challenged intraperitoneally (i.p.) with 50 μg OVA containing 2 mg Alum (Thermo Scientific) as adjuvant. In some experiments, 250 μg blocking anti-PD-L2 (Clone mAb 3.2, Leinco Technologies) or 250 μg IgG1 isotype control antibodies (BioXCell) was administered i.p. on days −10, −9, and −8 into WT mice as described before[16]. To measure airway inflammation and tolerance, mice were further challenged in some experiments with 50 μg OVA i.n. on day 7, 8, and 9, and the selected readouts were performed on day 10. In some experiments, ethyl pyruvate (Sigma Aldrich, 100 mg/kg) was injected intraperitoneally on days 7, 8, and 9.

### Measure of AHR and collection of BAL fluid

Measurements of airway resistance was conducted using the Fine-Pointe RC system (Buxco Research Systems, Wilmington, NC), in which anaesthetized mice were mechanically ventilated as previously described[53]. Mice were sequentially challenged with aerosolized PBS (baseline) followed by increasing doses of methacholine (Sigma) ranging from 2.5 to 40 mg/mL. Maximum resistance was recorded during a 3-minute period following each challenge as we continuously computed lung resistance ($R_L$) by fitting flow, volume, and pressure to an equation of motion. In some experiments, the trachea was cannulated, and the lungs were flushed three times with 1 ml ice-cold PBS to collect the BAL fluid. The frequency of the different leukocyte populations was determined by flow cytometry.

### Tissue preparation

Spleen and thymus from Foxp3^GFP and PD-L2^KO Foxp3^GFP WT or PD-L2^KO mice were minced through a 70 μm cell strainer and red blood cells lysed. In some experiments, lungs were perfused with ice-cold PBS through the left ventricle of the heart, digested with 400U/ml Collagenase IV (Worthington) for one hour at 37 °C, minced through a 70 μm cell strainer, and red blood cells lysed. Single-cell suspensions were then used for the selected readouts.

### Cell proliferation

To assess cell proliferation, mice were euthanized on day 7 and $1 \times 10^6$ splenocytes were cultured with 0–250 μg/ml OVA-peptide 323–339 (Invivogen) for 3 or 4 days in cRPMi. cRPMi was prepared using RPMI 1640 growth medium (Gibco), 10% Fetal Bovine Serum (Promega scientific) and 100 U/ml Penicillin/Streptomycin. Prior to culture, splenocytes were labeled with 1 μM CellTrace Violet (CTV, Thermofisher) for 20 min and stained on day 4 with CD3 and CD4 to isolate proliferating T cells and measure CTV dilution. In some experiments, proliferation was measured following 3 days of culture, where 1μ Ci $^3$H thymidine (MP Biomedicals, Solon, OH) was then added overnight and the amount of thymidine incorporation was quantified from harvested cells with a scintillation counter (Beckton Dickinson).

### Flow cytometry

The following murine antibodies were used: PECy7 anti-mouse CD45 (30-F11), APCCy7 anti-mouse CD45 (30-F11), PerCPCy5.5 anti-mouse CD3 (17A2), FITC anti-mouse CD3 (17A2), BV421 anti-mouse CD4 (GK1.5), APCCy7 anti-mouse CD4 (GK1.5), PECy7 anti-mouse CD8 (53-6.7), APCCy7 anti-mouse CD25 (PC61), BV421 anti-mouse CD25 (PC61), BV510 anti-mouse CD25 (PC61), PE anti-mouse CD304 (Neuropilin-1, 3E12), APC anti-mouse CD62L (MEL-14), APCCy7 anti-mouse CD44 (IM7), PECy7 anti-mouse CD44 (IM7), PerCPCy5.5 anti-mouse TCR DO11.10 (KJ1-26), APCCy7 anti-mouse CD11c (N418), PerCPCy5.5 anti-mouse CD11c (N418), APC anti-mouse CD170 (SiglecF, S17007L), PECy7 anti-mouse Ly6G (1A8), BV510 anti-mouse A/I-E (M5/114.15.2), APC anti-mouse CD274 (PD-L1, 10F9G2), PECy7 anti-mouse CD273 (PD-L2, TY25), BV421 anti-mouse CD279 (PD-1, 29F1A12), FITC anti-mouse CD19 (6D5), APC anti-mouse Gr-1 (RB6-8C5), FITC anti-mouse Foxp3 (MF-14) were purchased from BioLegend. PE anti-mouse CD170 (SiglecF, E50-2440) was purchased from BD Biosciences. eFluor450 anti-mouse CD11b (M1/70) was purchased from Thermofisher. Intranuclear staining was performed using the Foxp3 Transcription Factor Staining Kit (Thermofisher) according to the manufacturer's instructions. In some experiments, BODIPY^493/503 (Thermofisher) was used according to the manufacturer's instructions. Live/dead fixable violet or aqua cell stain kits were used to exclude dead cells, used according to the manufacturer's instructions (Thermofisher) and CountBright absolute counting beads (Thermofisher) to calculate absolute cell numbers when indicated. Stained cells were analyzed on FACSCanto II and/or FACSARIA III systems (Supplementary Table 1). BD FACSDiva software v8.0.l was used for flow cytometry data acquisition, Flowjo

software (TreeStar) version 9, and GraphPad Prism software version 8 were used for data analysis.

## Cell sorting and Treg cultures

Live tTregs (CD3+CD4+CD25+Foxp3GFP+Nrp1+) and pTregs (CD3+CD4+CD25+Foxp3GFP+Nrp1−) were isolated from the spleens of Foxp3GFP and PD-L2KO Foxp3GFP mice. $0.2–1.5 \times 10^5$ cells were further stimulated in U-bottom 96-well plates with a 1:1 ratio of Dynabeads Mouse T-activator CD3/CD28 (Gibco), and in some experiments 20 ng/mL IL-2 (Biolegend) and 5 ng/mL TGF-β (ebioscience) in cRPMI. Cells were incubated for up to 4 days at 37 °C and analyzed for the selected readouts. In some experiments, 2 mM methyl pyruvate (Sigma) was added to the cultures. When indicated, CD3−CD4+CD44−CD62L+ naive T cells were isolated from BALB/c splenocytes. For experiments involving iTregs, $1–10 \times 10^5$ naive T cells were incubated in U-bottom 96-well plates or flat-bottom 48-well plates with a 1:1 ratio of Dynabeads Mouse T-activator CD3/CD28 (Thermofisher), 20 ng/mL IL-2 (Biolegend) and 5 ng/mL TGF-β (ebioscience) in cRPMI. When indicated, 5 μg/mL of rmPD-L2Fc or corresponding rmIgG1Fc control (both R&D Systems) were coated in the wells for 30–60 min at 37 C before culture. For the BMDC/Treg co-cultures, OVA-specific KJ126+CD4+CD44−CD62L+ naive T cells were isolated from DO11.10 splenocytes and used as described below. When indicated, $10^6$ WT Foxp3GFP iTregs were adoptively transferred to WT BALB/c or PD-L2KO mice.

## BMDC cell differentiation and adoptive transfer

Bone marrow cells were flushed from the tibia and femur of WT BALB/c or PD-L2KO mice. $2–4 \times 10^6$ cells were then cultured in 10 ml cRPMi supplemented with 40 ng/mL Granulocyte maturation colony-stimulating factor (GM-CSF) for 7 to 10 days in a 9-cm diameter tissue culture coated petri dishes. Loosely adherent cells were collected and considered as BMDCs as they were >90% CD11c+ based on FACS analysis. In some experiments, WT or PD-L2KO BMDCs were stimulated overnight with 50 ng/mL rmIL-4 (Biolegend) and the amount of PD-L2 expression was determined by flow cytometry. Cells were washed and resuspended in PBS prior to transferring $1 \times 10^6$ BMDCs intravenously (i.v.) into WT Foxp3GFP or PD-L2KO Foxp3GFP recipients. Spleens were then harvested on day 3 and the numbers of CD3+CD4+CD25+Foxp3GFP+Nrp1+/− Tregs quantified by flow cytometry.

## Treg Suppression Assay and BMDC co-cultures

For the suppression assay, $2.5 \times 10^4$ CD4+CD44−CD62L+ naive T cells were isolated from the spleens of BALB/c mice and co-cultured with live Foxp3GFP or PD-L2KO Foxp3GFP sorted splenic pTregs (CD3+CD4+CD25+Foxp3GFP+Nrp1−) at 1:2; 1:4, 1:8, and 1:16 Treg:T cell ratios in cRPMi as described above for 72 h. For the BMDC co-cultures, WT or PD-L2KO BMDCs were generated as described above. $4 \times 10^5$ naive KJ126+CD4+CD25+CD62L+ splenic T cells were then isolated and co-cultured with WT or PD-L2KO BMDCs at a 1:20 DC to T-cell ratio. Cells were cultured in cRPMi containing 20 ng/mL IL-2 (Biolegend) and 5 ng/mL TGF-β (ebioscience), 100 ng/ml OVA-peptide 323–339 (Invivogen) and T cells were stimulated with a 1:1 ratio of Dynabeads Mouse T-activator CD3/CD28 (Gibco) for 72 h.

## Measurement of mitochondrial function

The real-time oxygen consumption rate (OCR) was measured using a Seahorse Mini HS XF instrument (Agilent). Briefly, $1–1.5 \times 10^5$ FACS-sorted Foxp3GFP and PD-L2KO Foxp3GFP pTregs were incubated as described above for 24 h and iTregs were differentiated with or without PD-L2Fc. Cells were then transferred on a Seahorse XFp PDL-coated cell culture miniplate in FBS/Phenol red free pH7.4 Seahorse media supplemented with 1 mM pyruvate, 2 mM glutamine and 10 mM glucose. A Mito Stress Test assay was then performed (Agilent). Briefly following baseline measurements, 1 μM oligomycin, 2 μM FCCP and 0.5 μM rotenone/antimycinA were sequentially injected into the culture. Oxygen levels in the culture were measured in triplicates following each injection and OCR was computed. Seahorse data was analyzed using the Seahorse data analytics platform (https://seahorseanalytics.agilent.com). For analysis of mitochondrial mass, cells were incubated with 200 nM MitoTrackerDR fluorescent probe (Thermofisher) for 30 min at 37 °C and accumulation of the probe was detected in the APC channel by flow cytometry.

## RNA-sequencing and TSDR methylation analysis

Transcriptomic analysis was performed as described previously[54]. Briefly, cultured pTregs were recovered, lysed in RLT buffer (Qiagen) and RNA was extracted using the MicroRNeasy kit (Qiagen). For each sample, a total of 10 pg of RNA was used to generate cDNA (SMARTer Ultra Low Input RNA v3 kit, Clontech) for library preparation. Samples were then amplified and sequenced on a NextSeq 500 system (Illumina) where on average 30 million reads were generated from each sample. Raw reads were then further processed on Partek Genomics Suite software, version 7.0; Partek Inc. Briefly, raw reads were aligned by STAR – 2.6.1d with mouse reference index mm10 and GENECODE M21 annotations. Aligned reads were further quantified and normalized using the upper quartile method and differential analysis by GSA. Transcripts showing an average normalized count below 1 were removed from the analysis, as were genes showing cumulative normalized counts below 10. For the methylation analysis, Foxp3 DNA methylation was performed by bisulfite pyrosequencing by EpigenDX as described previously[55]. Briefly, Foxp3GFP and PD-L2KO Foxp3GFP pTregs were isolated and cultured as described above, and iTregs were differentiated with or without PD-L2Fc. Genomic DNA was then extracted from cultures, followed by bisulfite conversion, purification, cloning, and the degree of methylation of CpG#19, CpG#20, CpG#21, and CpG#22 within the TSDR region was determined by bisulfite sequencing.

## Cytokine measurement

When indicated, culture supernatants were collected and cytokines measured using the LEGENDplex™ Mouse Th2 Panel or the mouse IL-2 ELISA MAX kit (Biolegend) according to the manufacturer's instructions.

## Statistical analysis

Data presented in the Figures are from a single (non-pooled) representative experiment, with the appropriate sample size and number of repeats indicated in the legends. A two-tailed Student's $t$ test for unpaired data was applied for comparisons between two groups, except for multi-group comparisons where one-way analysis of variance tests were used. All tests were performed using Prism Software (GraphPad Software Inc.).

## Reporting summary

Further information on research design is available in the Nature Research Reporting Summary linked to this article.

## Data availability

The RNA-seq data have been deposited in the Genbank database under the GEO accession code GSE210360. All data are included in the Supplemental Information or available from the authors upon reasonable requests, as are unique reagents used in this Article. The raw numbers for charts and graphs are available in the Source Data file whenever possible. Source data are provided with this paper.

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

## Acknowledgements
This article was financially supported by National Institutes of Health Public Health Service grants R01 HL144790, R01 HL151493, R01 AI145813 and R01 HL159804 (O.A.) and P01 AI56299 (A.H.S.). We are grateful to USC Libraries Bioinformatics Service for assisting with data analysis, in particular Dr. Yibu Chen and Dr. Meng Li. The bioinformatics software and computing resources used in the analysis are funded by the USC Office of Research and the Norris Medical Library.

## Author contributions
B.P.H. designed and performed experiments, analyzed, interpreted the results, and wrote the manuscript. D.G.H, E.H., J.D.P., and P.S-J. helped perform experiments and provided animal husbandry for experiments. A.H.S. provided reagents and critically reviewed the manuscript. O.A. supervised, designed the experiments, interpreted the data, and critically reviewed the manuscript.

## Competing interests
Authors have no competing interests to declare except for A.H.S. A.H.S. has patents/pending royalties on the PD-1 pathway from Roche and Novartis. A.H.S. is on advisory boards for Surface Oncology, Elstar, SQZ Biotechnologies, Elpiscience, Selecta, Bicara and Monopteros, GlaxoSmithKline, and Janssen. A.H.S. has received research funding from Novartis, Roche, UCB, Ipsen, Merck, and AbbVie unrelated to this project.
