## [Peer Review File · Nature Communications]

PD-L2 controls peripherally induced regulatory T cells by maintaining metabolic activity and Foxp3 stabilityREVIEWER COMMENTS

Reviewer #1 (Remarks to the Author):

In this paper by Hurrell et. Al, the authors explore the role of PD-L2 in the control of responses against the model antigen OVA. They make use of a model of airway exposure to OVA together with mice defective for PD-L2 expression. Previously, the authors have shown that loss of PD-L2 leads to an exuberant inflammatory response to OVA exposure in the lungs. Here they explore the effect of PD-L2 deletion on the number, phenotype, and function of Foxp3 Treg cells. Importantly, they demonstrate that in response to lung OVA exposure, there is a reduction in the number of Nrp1- Tregs in the lung when the PD-L2 is deleted. Furthermore, they make use of IL-4 treated BMDC defective for PD-L2 to clearly demonstrate in vitro that PD-L2 can be necessary for inducible Treg generation.

Given the relative lack of data on PD-L2 relative to PD-L1, and the questions that have persisted about the ability of PD-L2 to act as a co-inhibitory ligand for PD-1 (much like PD-L1), the data shown here advance the literature. Nevertheless, there are numerous weaknesses in the paper that lower my enthusiasm for the work:

1) The abstract and discussion highlight the importance of this work in demonstrating a role for PD-L2 in establishing immune tolerance to inhaled antigens. This is important because their previous work already demonstrated that PD-L2^{-/-} mice respond more aggressively to airway antigen exposure. Unfortunately, no true investigation of immune tolerance is offered here. There is no demonstration that isolated conventional CD4 T cells from either spleen or lung of PD-L2-deficient mice are less tolerant of antigen on recall. The proliferation assay and IL-2 production in Figure 1 are performed in the presence of APC that either express or lack PD-L2 as well as in the presence of altered Treg populations. More importantly, tolerance experiments should include a no-antigen control, and this was not provided here. In summary, all of the lung-related functional studies are simply repetitive of their previous published results. The authors have not demonstrated a role for PD-L2 in peripheral immune tolerance induction

2) A lack of appropriate controls is a common theme in this work. Figure 2 illustrates the effects of WT BMDC administration on Treg cell homeostasis in the PD-L2^{-/-} hosts. But PD-L2^{-/-} BMDC are not shown for their effects. Furthermore, only PD-L2^{-/-} recipients are tested, while effects on WT hosts are not considered here. Finally, BMDC are sorted based on PD-L2 expression, and other DC characteristics may correlate to PD-L2 expression level—is the effect of these transferred BMDC blocked with anti-PD-L2 mAb? Is increased Treg number associated with normalized function?

3) The relationship of CNS2 demethylation to pTreg differentiation and function are not entirely clear here. Generally, it is thought that CNS2 demethylation in the thymus is key to stable thymic Treg Foxp3 expression. On the other hand, Foxp3 expression on Nrp1- Tregs is thought to be Tgfb1-dependent and occurs without CNS2 demethylation. Anergy-derived pTreg cells expressing Nrp1 have been shown to demethylate the CNS2 region. In this paper, there is a focus on Nrp1- apparently pTreg cells or iTregs generated in vitro in the presence of Tgfb1. Furthermore, there is discussion of Treg stability when only modest differences in Foxp3 MFI are shown. The authors should have tested for stability and ex-Foxp3 generation following transfer of Tregs for PD-L2^{-/-} mice to wild-type hosts. A rigorous investigation of this problem would also include a phenotypic and function examination of thymic Tregs from WT and PD-L2^{-/-} hosts. Like the peripheral Nrp1+ Tregs, is thymic Treg differentiation normal in the absence of PD-L2? Is this really a peripheral phenotype?

Reviewer #2 (Remarks to the Author):

Dr. Akbari and colleagues studied a role of PD-L2 in peripheral Treg biology and pathology. The topic is interesting. However, the work suffers from several major weaknesses.

1. Role of DC PD-L2: It seems that the authors attempted to demonstrate a role of DC PD-L2 in Tregs. However, they touched this without furthering their studies.
2. IL-10 and FOXP3 in Tregs: It seems that the authors attempted to demonstrate roles of PD-L2 in affecting IL-10 and FOXP3 in Tregs. However, they touched this without furthering their studies. It remains unknown if and how FOXP3 and/or IL-10 contribute to respiratory tolerance. Similarly, they touched mitochondria activities without furthering their studies. In summary, the mechanistic and functional connection among PD-L2, iTreg (IL-10 and FOXP3), and iTreg metabolism is not cohesively established in homeostatic situation or respiratory tolerance model.

Reviewer #3 (Remarks to the Author):

Hurrell et al reported in this manuscript that PD-L2 controls pTregs by maintaining metabolic activity and Foxp3 stability and highlighted the important role of the PD-1/PD-L2 axis in respiratory tolerance. They showed that PD-L2 KO mice failed to induce immune tolerance and a lower number of systemic Foxp3+ Tregs under homeostatic conditions and could restore these numbers by adoptively transferring PD-L2 high dendritic cells. Mechanistically, they showed PD-L2 KO Tregs decreased TCA cycle, defected in mitochondrial function and ATP production, and responded to pyruvate treatment for partially restoring IL-10 production. The reported results are interesting and may improve mucosal immunotherapy by including metabolic or costimulatory modulators to modify pTreg suppressive function and establish immune tolerance. However, the current manuscript does not provide sufficient evidence to delineate the mechanisms by which PD-L2 differentially regulates the methylation at the Foxp3 TSDR region and controls pTreg metabolic activity and reprograms.

Major points

1. Fig. 2 F and G. The adoptively transfer of BMDCs expressing high levels of PD-L2 into PD-L2-KO recipient mice restored the number of Foxp3+ Tregs at day 3 after transfer. Since the manuscript is focusing on the impact of PDL2 on pTreg, the authors should show the effect of the adoptive DC transfers on both pTreg vs tTreg. Also helpful if the authors show the effect on pTreg and tTreg numbers in WT recipients for comparison.
2. The authors primarily used cell count to quantitate differences in cell populations. This is viewed as a strength since frequency could be technically misleading. Still, it would be insightful to have the frequencies of cell populations (i.e. % relative to total CD45) provided in supplemental materials.
3. The authors used Nrp1 surface marker for FACS purification of tTreg and pTreg populations for in vitro assays described in Fig. 3. The figure legend states that the cells were stimulated with aCD3/CD28 beads plus IL-2 and TGF-B as a prerequisite to evaluating IL-10 production by the tTregs or pTregs. Since Nrp1 is a TGF-B receptor, the possible impact of anti-Nrp1 Ab used for FACS sorting should not be overlooked. The authors should confirm the observed functional differences in pTreg vs tTreg are not due to discrepancies in TGF-B stimulation due to the Nrp1 Ab, or inform the reader that this was a limitation of the experiment since there are no reliable markers of pTreg or tTreg in vivo.

4. Fig. 4. In the manuscript authors argued that PD-L2 restricted DNA methylation at the Foxp3 TSDR region in pTregs cells from PD-L2-KO mice. If the authors want to make this statement, then they need to thoroughly address what is the mechanism that PD-L2 regulates DNA methylation at the Foxp3 TSDR region. Does expression of PD-L2 in pTregs from PD-L2-KO mice regulate DNA de-methylation at the Foxp3 TSDR region?
5. Fig. 5D-G showed that knockout of PD-L2 in pTreg differentially regulated genes involved in FAO, AA degradation and TCA cycle. The authors should also present genes in glycolysis and pentose phosphate pathway (PPP) pathways. Does KO of PD-L2 reprogram Treg metabolism? How can authors explain why pyruvate treatment of PD-L2-KO Tregs were able to partially restore IL10 production in vitro (Fig. 5N&O)? Can pyruvate restore the respiratory tolerance in PD-L2-KO mice?
6. Fig. 6. In fig 6B&C, authors showed that the frequencies of pTreg within total Tregs cells in PD-L2-KO mice is decreased compared to WT mice. How PD-L2 controls pTreg numbers, inhibit proliferation or induce apoptosis? In Fig. 6G&H, authors showed that WT and PD-L2-KO BMDCs both induced Foxp3 and the difference is minor (Fig. 6H). Does this indicate the difference of frequencies of pTreg in PD-L2-KO from WT mice in Fig. 6B&C? Authors should address this issue.
7. In Statistical analysis section on page 11 the authors stated that “experiments were repeated at least two times (n=5-10) and data are shown as a representative experiment”. The figure legends stated n=3 or n=5, so I am confused whether the n=5-10 is referring to the sum of two experiments, or just a single experiment? The sample size and statistical analysis should be clearly described to help the reader understand how the conclusions were made. Due to the confusion of this, I strongly suggest the authors present their quantitative data as dot plots with the average and the appropriate error bars indicated which provides clarity and transparency and allow the audience to more critically evaluate their work. The authors should indicate how many mice were used in each groups and how many repeats in figure legends.

Minor points

1. On page 5 to 6, the sentence (line 111 to line 114) - As it is currently written, it feels incomplete when reading it. The authors should extend the sentence to describe the effects of the phosphorylation.
2. There are a few typos throughout. In the introduction, line 98, neuropilin 1 is misspelled as neutropilin 1.
3. Figure 1C, the y axis label, CD45 + is subscripted and should be corrected to superscript
4. Line 356 should be Figure 2G not Figure 1G

We thank the Reviewers for their constructive comments. We specially believe the Reviewers' criticism are reasonable and the additional data requested further strengthens our manuscript. We have performed numerous additional experiments and **addressed all concerns raised by the Reviewers**. These include experiments assessing the proliferation of CD4⁺ T-cells in the OVA-tolerance model, adoptive transfer experiments, novel methylation assays, bioenergetic assays, further analysis of our RNAseq data and in vitro experiments using PD-L2Fc to mimic PD1/PD-L2 interactions. Please see the detailed point by point response below, referencing the relevant sections of the manuscript. In the marked copy, these modifications have been highlighted in **green**.

Reviewer #1

In this paper by Hurrell et. Al, the authors explore the role of PD-L2 in the control of responses against the model antigen OVA. They make use of a model of airway exposure to OVA together with mice defective for PD-L2 expression. Previously, the authors have shown that loss of PD-L2 leads to an exuberant inflammatory response to OVA exposure in the lungs. Here they explore the effect of PD-L2 deletion on the number, phenotype, and function of Foxp3 Treg cells. Importantly, they demonstrate that in response to lung OVA exposure, there is a reduction in the number of Nrp1- Tregs in the lung when the PD-L2 is deleted. Furthermore, they make use of IL-4 treated BMDC defective for PD-L2 to clearly demonstrate in vitro that PD-L2 can be necessary for inducible Treg generation.

Given the relative lack of data on PD-L2 relative to PD-L1, and the questions that have persisted about the ability of PD-L2 to act as a co-inhibitory ligand for PD-1 (much like PD-L1), the data shown here advance the literature. Nevertheless, there are numerous weaknesses in the paper that lower my enthusiasm for the work:

We thank the Reviewer for the constructive comments, please find below the response to the specific comments:

1) The abstract and discussion highlight the importance of this work in demonstrating a role for PD-L2 in establishing immune tolerance to inhaled antigens. This is important because their previous work already demonstrated that PD-L2^{-/-} mice respond more aggressively to airway antigen exposure. Unfortunately, no true investigation of immune tolerance is offered here. There is no demonstration that isolated conventional CD4 T cells from either spleen or lung of PD-L2-deficient mice are less tolerant of antigen on recall. The proliferation assay and IL-2 production in Figure 1 are performed in the presence of APC that either express or lack PD-L2 as well as in the presence of altered Treg populations. More importantly, tolerance experiments should include a no-antigen control, and this was not provided here. In summary, all of the lung-related functional studies are simply repetitive of their previous published results. The authors have not demonstrated a role for PD-L2 in peripheral immune tolerance induction.

In our previous publication, mice were first sensitized with OVA in alum intraperitoneally and starting on day 8 challenged on three consecutive days with OVA intranasally. The Reviewer is correct, as PD-L2^{KO} showed more inflammation compared to controls. However importantly, the proliferation of OVA-specific T-cells was unaffected in this model (PMID: 19741598 and 24752301). In the tolerance model used in our manuscript, mice are first challenged with OVA intranasally on three consecutive days, a week prior sensitization with OVA in alum. Although we agree that there may be some inevitable similarities in the experiments performed between our previous publication and the current, the differences in the airway inflammation model used make our described findings novel and not simply repetitive of our previous published results.

In particular we have now addressed the comment on the assessment of immune tolerance by performing a new set of experiments. Briefly, we challenged WT and PD-L2^{KO} mice intranasally with OVA on days -10, -9 and -8. Mice were then sensitized on day 0 with OVA in alum intraperitoneally and on day 7 we compared the number of T-cells responsive to OVA challenge as described previously (PMID: 24752301). Splenocytes were isolated and labeled with CellTrace Violet (CTV) before culture with or without OVA protein for 96 hours. Importantly, we found that the proliferation of CD3⁺CD4⁺ T-cells in response to OVA was significantly higher in PD-L2^{KO} tolerized mice as compared to WT tolerized mice. As requested, we have included a no-antigen control in the experiment, as the new data can now be found in **Figure 1D-E**. The data clearly demonstrates the absence of proliferation in the absence of OVA in vivo, as the observed effects are not due to autoreactive T-cells. Importantly, we also did not see proliferation in the absence of OVA in vitro, suggesting that the proliferating T-cells are OVA-specific. We always add no-antigen controls in our airway hyperreactivity experiments and have included it in the new version of the Figure (**Figure 1F-H**). We therefore believe that our data demonstrates a tolerance reduction in PD-L2^{KO} mice, since CD4⁺ T cells from PD-L2^{KO} mice are less tolerant of antigen on recall.

2) A lack of appropriate controls is a common theme in this work. Figure 2 illustrates the effects of WT BMDC administration on Treg cell homeostasis in the PD-L2^{-/-} hosts. But PD-L2^{-/-} BMDC are not shown for their effects. Furthermore, only PD-L2^{-/-} recipients are tested, while effects on WT hosts are not considered here. Finally, BMDC are sorted based on PD-L2 expression, and other DC characteristics may correlate to PD-L2 expression level—is the effect of these transferred BMDC blocked with anti-PD-L2 mAb? Is increased Treg number associated with normalized function?

This is an excellent point raised by the Reviewer and we have now performed two additional adoptive transfer experiments. We first assessed the effects of the adoptive transfer of PD-L2^{low} and PD-L2^{high} BMDCs on total Tregs in both WT and PD-L2^{KO} recipients. The data clearly demonstrates that neither PD-L2^{low} nor PD-L2^{high} BMDCs significantly affects splenic Treg numbers in WT recipients, whereas as shown already in the first version of the manuscript, only PD-L2^{high} BMDCs restored the numbers of splenic Tregs in PD-L2^{KO} mice to levels comparable to WT mice (**Figure 3I**). Next, we adoptively transferred both WT or PD-L2^{KO} BMDCs to PD-L2^{KO} hosts to characterize the effects of PD-L2 on total splenic Tregs. Remarkably, only WT BMDCs restored the numbers of splenic Tregs, confirming that the observed effects are PD-L2 dependent and not due to other DC characteristics (**Figure 3J**). More importantly, we further characterized restored Tregs as requested by Reviewer 3 and found that BMDCs affected pTreg and not tTreg numbers (**Figure 3K-L**), a finding we confirmed by seeing no effect of PD-L2 on Treg progenitors in the thymus (**Figure 3M**). We believe these new data demonstrate that PD-L2 on BMDCs is able to control Treg – in particular pTreg – numbers in the PD-L2^{KO} host.

3) The relationship of CNS2 demethylation to pTreg differentiation and function are not entirely clear here. Generally, it is thought that CNS2 demethylation in the thymus is key to stable thymic Treg Foxp3 expression. On the other hand, Foxp3 expression on Nrp1- Tregs is thought to be Tgfb1-dependent and occurs without CNS2 demethylation. Anergy-derived pTreg cells expressing Nrp1 have been shown to demethylate the CNS2 region. In this paper, there is a focus on Nrp1- apparently pTreg cells or iTregs generated in vitro in the presence of Tgfb1. Furthermore, there is discussion of Treg stability when only modest differences in Foxp3 MFI are shown. The authors should have tested for stability and ex-Foxp3 generation following transfer of Tregs for PD-L2^{-/-} mice to wild-type hosts.

We have now performed multiple new experiments to address the Reviewer's comment. It is important to point out that we did not find expression of PD-L2 by either splenic or lung Tregs, neither at steady state nor following inflammation induced by OVA, as PD-L2 is mainly expressed on dendritic cells (**Figure 1A-B**). To directly measure the effects of PD-L2 on Treg development, maintenance and function, we now performed new in vitro experiments using a PD-L2Fc to mimic PD-1/PD-L2 interactions. The data can now be found in an entirely new **Figure 2**. Our novel findings clearly demonstrate that PD-L2 increases the number of induced Tregs and production of IL-10. Furthermore, and in confirmation of our bioenergetic studies presented in **Figure 5**, PD-L2 increased OXPHOS and ATP production in induced Tregs. Importantly, we found that PD-L2 increased the stability of Foxp3, which was associated with a lower frequency of Foxp3 methylation. As requested, we have now performed new experiments to better assess Foxp3 stability in vivo. Since Tregs do not express PD-L2, we adoptively transferred WT Foxp3^{GFP} induced Tregs to either WT or PD-L2^{KO} recipients (non Foxp3^{GFP}) and measured the numbers of Foxp3^{GFP} Tregs in both the spleen and the lungs, as a measure of Foxp3 stability. Remarkably, we found significantly less GFP in PD-L2^{KO} hosts compared to controls in both spleen and lungs, suggesting for a less stable Foxp3 in the absence of PD-L2 in the host. The new data can now be found in **Figure 4K-M**. Together with our new findings now presented in Figure 2, we believe this latest experiment confirms a pivotal role for PD-L2 in induced Treg development and stability.

A rigorous investigation of this problem would also include a phenotypic and function examination of thymic Tregs from WT and PD-L2^{-/-} hosts. Like the peripheral Nrp1+ Tregs, is thymic Treg differentiation normal in the absence of PD-L2? Is this really a peripheral phenotype?

This is a very important point. Since we observed an effect of PD-L2 on the development of peripheral Tregs, we now show that PD-L2 does not affect thymic Treg development in two separate experiments. Tregs are generated via two distinct developmental programs including the CD25⁺ Treg cell progenitors (CD25⁺ TregP cells) and the Foxp3^{lo} Treg cell progenitors (Foxp3^{lo} TregP cells). Importantly, we did not find a difference in Treg progenitors in PD-L2^{KO} mice compared to controls at steady state, suggesting that thymic Treg development is normal in the absence of PD-L2 (**Figure 3A-B**). We also approached this point in the novel adoptive transfer experiments requested by the Reviewer. Although we found differences in the numbers of splenic pTregs in mice that received WT BMDCs, it did not affect the numbers of tTregs. More importantly, it did not modulate the frequencies of Treg progenitors in the thymus (**Figure 3M**). Together, these new sets of experiment provide complementing evidence that PD-L2 affects the development of Tregs in the periphery rather than that of natural Tregs.

Reviewer #2

Dr. Akbari and colleagues studied a role of PD-L2 in peripheral Treg biology and pathology. The topic is interesting. However, the work suffers from several major weaknesses.

Based on the comments by all Reviewers, we have now performed multiple key additional experiments that we believe address the concerns raised by the Reviewer. Please find below our response to the Reviewer's comments.:

- 1. Role of DC PD-L2: It seems that the authors attempted to demonstrate a role of DC PD-L2 in Tregs. However, they touched this without furthering their studies.*
- 2. IL-10 and FOXP3 in Tregs: It seems that the authors attempted to demonstrate roles of PD-L2 in affecting IL-10 and FOXP3 in Tregs. However, they touched this without furthering their studies. It remains unknown if and how FOXP3 and/or IL-10 contribute to respiratory tolerance. Similarly,*

they touched mitochondria activities without furthering their studies. In summary, the mechanistic and functional connection among PD-L2, iTreg (IL-10 and FOXP3), and iTreg metabolism is not cohesively established in homeostatic situation or respiratory tolerance model.

1) As suggested by Reviewers 1 and 3, we have extensively clarified the role of DC PD-L2 in the development of peripheral Tregs. We first assessed the effects of the adoptive transfer of PD-L2^{low} and PD-L2^{high} BMDCs on total Tregs in both WT and PD-L2^{KO} recipients. The data clearly demonstrates that neither PD-L2^{low} nor PD-L2^{high} BMDCs significantly affects splenic Treg numbers in WT recipients, whereas as shown already in the first version of the manuscript, only PD-L2^{high} BMDCs restored the numbers of splenic Tregs in PD-L2^{KO} mice to levels comparable to WT mice (**Figure 3I**). Next, we adoptively transferred both WT and PD-L2^{KO} BMDCs to PD-L2^{KO} hosts to characterize the effects of PD-L2 on total splenic Tregs. Remarkably, only WT BMDCs restored the numbers of splenic Tregs, confirming that the observed effects are PD-L2 dependent and not due to other DC characteristics (**Figure 3J**). More importantly, we further characterized restored Tregs as requested by Reviewer 3 and found that BMDCs affected pTreg and not tTreg numbers (**Figure 3K-L**), a finding we confirmed by seeing no effect of PD-L2 Treg progenitors in the thymus (**Figure 3M**). We believe these new data demonstrate that PD-L2 on BMDCs is able to control Treg – in particular pTreg – numbers in the PD-L2^{KO} host.

2) We would like to point out that the role of IL-10 in respiratory tolerance was previously reported (PMID:11477409), as induction of tolerance was described following adoptive transfer of Tregs (PMID:17876110). As requested by Reviewer 1, we have expanded our studies on the role of PD-L2 in affecting Treg development, maintenance and function. In particular, we now performed new in vitro experiments using a PD-L2Fc to mimic PD-1/PD-L2 interactions, and the data can now be found in an entirely new **Figure 2**. Our novel findings clearly demonstrate that PD-L2 increases the number of induced Tregs and production of IL-10. Furthermore, and in confirmation of our bioenergetic studies presented in **Figure 5**, PD-L2 increased OXPHOS and ATP production in induced Tregs. Importantly, we found that PD-L2 increased the stability of Foxp3, which was associated with a lower frequency of Foxp3 methylation. Furthermore, we have now performed new experiments to better assess Foxp3 stability in vivo. Since Tregs do not express PD-L2, we adoptively transferred WT Foxp3^{GFP} induced Tregs to either WT or PD-L2^{KO} recipients (non Foxp3^{GFP}) and measured the numbers of Foxp3^{GFP} Tregs in both the spleen and the lungs, as a measure of Foxp3 stability. Remarkably, we found significantly less GFP in PD-L2^{KO} hosts compared to controls, suggesting for a less stable Foxp3 in the absence of PD-L2 in the host. The new data can now be found in **Figure 4K-M**. Together with our new findings now presented in Figure 2, we believe this latest experiment confirms a pivotal role for PD-L2 in induced Treg development and stability.

3) As requested by Reviewer 3, we have now added glycolysis (**Figure 5F**) and Pentose Phosphate Pathway (**Figure 5G**) genes to our metabolic analysis. Surprisingly, we did not see any significant modulation of either pathways, suggesting that the absence of PD-L2 does not induce a reprogramming of Treg metabolism. As shown in our novel **Figure 2**, exogenous PD-L2 interactions with Treg rather induces the use of FAO as an energy source, as we did not see differences in ECAR at basal levels. Furthermore, we have now performed additional experiments to address whether in vivo treatment of mice with pyruvate could restore the respiratory tolerance in PD-L2^{KO} mice, and the new data can now be found in **Figure 6K-M**. Although sham-treated tolerized PD-L2^{KO} mice induced a substantial inflammatory response in response to OVA challenge, treatment with pyruvate intraperitoneally on 3 consecutive days remarkably decreased both airway hyperreactivity and BAL inflammatory infiltrates to levels compared to tolerized WT mice.

4.) We have further performed multiple other experiments and adapted the Figures and the text accordingly. We specially believe the Reviewers' criticism were reasonable and the additional data requested further strengthened our manuscript.

Reviewer #3

Hurrell et al reported in this manuscript that PD-L2 controls pTregs by maintaining metabolic activity and Foxp3 stability and highlighted the important role of the PD-1/PD-L2 axis in respiratory tolerance. They showed that PD-L2 KO mice failed to induce immune tolerance and a lower number of systemic Foxp3+ Tregs under homeostatic conditions and could restore these numbers by adoptively transferring PD-L2 high dendritic cells. Mechanistically, they showed PD-L2 KO Tregs decreased TCA cycle, defected in mitochondrial function and ATP production, and responded to pyruvate treatment for partially restoring IL-10 production. The reported results are interesting and may improve mucosal immunotherapy by including metabolic or costimulatory modulators to modify pTreg suppressive function and establish immune tolerance. However, the current manuscript does not provide sufficient evidence to delineate the mechanisms by which PD-L2 differentially regulates the methylation at the Foxp3 TSDR region and controls pTreg metabolic activity and reprograms.

We thank the Reviewer for the constructive comments. We have now performed all the suggested experiments – including others – and believe it has further strengthened our manuscript. Please find our detailed response below.

Major points

1. Fig. 2 F and G. The adoptively transfer of BMDCs expressing high levels of PD-L2 into PD-L2-KO recipient mice restored the number of Foxp3+ Tregs at day 3 after transfer. Since the manuscript is focusing on the impact of PDL2 on pTreg, the authors should show the effect of the adoptive DC transfers on both pTreg vs tTreg. Also helpful if the authors show the effect on pTreg and tTreg numbers in WT recipients for comparison.

This is an excellent point also raised by Reviewer 1. As mentioned above, we have now performed a new set of experiments and the data can be found in **Figure 3**. Briefly, we first assessed the effects of the adoptive transfer of PD-L2^{low} and PD-L2^{high} BMDCs on total Tregs in both WT and PD-L2^{KO} recipients. The data clearly demonstrates that neither PD-L2^{low} nor PD-L2^{high} BMDCs significantly affects splenic Treg numbers in WT recipients, whereas as shown already in the first version of the manuscript, only PD-L2^{high} BMDCs restored the numbers of splenic Tregs in PD-L2^{KO} mice to levels comparable to WT mice (**Figure 3I**). Next, we adoptively transferred both WT and PD-L2^{KO} BMDCs to characterize the effects of PD-L2 on total splenic Tregs. Remarkably, only WT BMDCs restored the numbers of splenic Tregs, confirming that the observed effects are PD-L2 dependent and not due to other DC characteristics (**Figure 3J**). More importantly and as requested, we further characterized restored Tregs and found that BMDCs affected pTreg and not tTreg numbers (**Figure 3K-L**), a finding we confirmed by seeing no effect of PD-L2 on Treg progenitors in the thymus (**Figure 3M**). We believe these new data demonstrate that PD-L2 on BMDCs is able to control the numbers of Tregs, and in particular that of peripherally induced Tregs.

2. The authors primarily used cell count to quantitate differences in cell populations. This is viewed as a strength since frequency could be technically misleading. Still, it would be insightful to have

the frequencies of cell populations (i.e. % relative to total CD45) provided in supplemental materials.

We agree with the Reviewer and we have now added in **Supplementary Figure 3C-D** the requested data presented in % relative to CD3 for the major findings of the manuscript.

3. The authors used *Nrp1* surface marker for FACS purification of *tTreg* and *pTreg* populations for *in vitro* assays described in Fig. 3. The figure legend states that the cells were stimulated with *aCD3/CD28* beads plus *IL-2* and *TGF-B* as a prerequisite to evaluating *IL-10* production by the *tTregs* or *pTregs*. Since *Nrp1* is a *TGF-B* receptor, the possible impact of anti-*Nrp1* Ab used for FACS sorting should not be overlooked. The authors should confirm the observed functional differences in *pTreg* vs *tTreg* are not due to discrepancies in *TGF-B* stimulation due to the *Nrp1* Ab, or inform the reader that this was a limitation of the experiment since there are no reliable markers of *pTreg* or *tTreg* *in vivo*.

We agree with the Reviewer and we have now added a statement in the discussion regarding this issue.

4. Fig. 4. In the manuscript authors argued that PD-L2 restricted DNA methylation at the *Foxp3* TSDR region in *pTregs* cells from PD-L2-KO mice. If the authors want to make this statement, then they need to thoroughly address what is the mechanism that PD-L2 regulates DNA methylation at the *Foxp3* TSDR region. Does expression of PD-L2 in *pTregs* from PD-L2-KO mice regulate DNA de-methylation at the *Foxp3* TSDR region?

It is important to point out that we did not find expression of PD-L2 by either splenic or lung Tregs, neither at steady state nor following inflammation induced by OVA, as PD-L2 is mainly expressed on dendritic cells (**Figure 1A-B**). Whereas PD-L1 is highly expressed and induced in CD4⁺, CD8⁺ T-cells and Foxp3⁺ Tregs, PD-L2 expression is absent from CD3⁺ T-cells and limited to certain types of antigen presenting cells upon specific stimulation. We apologize for not being clear and have now clarified this statement in the manuscript. To directly measure the effects of PD-L2 on Treg development, maintenance and function, we now performed new *in vitro* experiments using a PD-L2Fc to mimic PD-1/PD-L2 interactions. The data can now be found in an entirely new **Figure 2**. Our novel findings clearly demonstrate that exogenous PD-L2 increases the number of induced Tregs and production of *IL-10*. Furthermore, and in confirmation of our bioenergetic studies presented in **Figure 5**, PD-L2 increased OXPHOS and ATP production in induced Tregs. Importantly, we found that PD-L2 increased the stability of *Foxp3*, which was associated with a lower frequency of *Foxp3* methylation. As requested by Reviewer 1, we now performed experiments to measure Treg stability *in vivo* whereby WT induced Tregs were adoptively transferred to WT or PD-L2^{KO} hosts. We found that the numbers of transferred Tregs were decreased 3 days post transfer in the absence of PD-L2 in the host compared to controls, suggesting that *Foxp3* stability is affected in the absence of PD-1/PD-L2 interactions *in vivo* (**Figure 4K-M**). Together, these new findings provide complementing evidence that exogenous PD-L2 binding to Tregs regulates Treg stability and function, rather than PD-L2 expressed on Tregs.

5. Fig. 5D-G showed that knockout of PD-L2 in *pTreg* differentially regulated genes involved in FAO, AA degradation and TCA cycle. The authors should also present genes in glycolysis and pentose phosphate pathway (PPP) pathways. Does KO of PD-L2 reprogram Treg metabolism? How can authors explain why pyruvate treatment of PD-L2-KO Tregs were able to partially restore *IL10* production *in vitro* (Fig. 5N&O)? Can pyruvate restore the respiratory tolerance in PD-L2-KO mice?

We agree with the Reviewer and have now added glycolysis (**Figure 5F**) and Pentose Phosphate Pathway (**Figure 5G**) genes to the Figure. Surprisingly, we did not see any significant modulation of either pathways, suggesting that the absence of PD-L2 does not induce a reprogramming of Treg metabolism. As shown in our novel **Figure 2**, exogenous PD-L2 interactions with Treg rather induces the use of FAO as an energy source, as we did not see differences in ECAR at basal levels. As requested, we have now performed additional experiments to address whether in vivo treatment of mice with pyruvate could restore the respiratory tolerance in PD-L2^{KO} mice, and the new data can now be found in **Figure 6K-M**. Although sham-treated tolerized PD-L2^{KO} mice induced a substantial inflammatory response in response to OVA challenge, treatment with pyruvate intraperitoneally on 3 consecutive days remarkably decreased both airway hyperreactivity and BAL inflammatory infiltrates to levels compared to tolerized WT mice. We have recently reported that exogenous pyruvate treatment similarly restores ILC2 effector function (PMID: 31738991). We hypothesize that fatty acids are used during Treg activation as energy substrates for FAO and OXPHOS and we used pyruvate as an energy metabolite that can substitute FA-derived acetyl-CoA to directly fuel OXPHOS and promote mitochondrial respiration. We apologize for not being clear in the manuscript and have adapted the text accordingly.

6. Fig. 6. In fig 6B&C, authors showed that the frequencies of pTreg within total Tregs cells in PD-L2-KO mice is decreased compared to WT mice. How PD-L2 controls pTreg numbers, inhibit proliferation or induce apoptosis? In Fig. 6G&H, authors showed that WT and PD-L2-KO BMDCs both induced Foxp3 and the difference is minor (Fig. 6H). Does this indicate the difference of frequencies of pTreg in PD-L2-KO from WT mice in Fig. 6B&C? Authors should address this issue.

As previously discussed, we have now performed additional in vitro experiment using a PD-L2Fc to mimic PD-1/PD-L2 interactions during iTreg induction (**Figure 2**). Our findings clearly show that exogenous PD-L2 interactions favor induced Treg development, as evidenced by the higher numbers of iTregs in the PD-L2Fc condition compared to corresponding IgG1Fc controls. This does however not rule out an effect of PD-L2 in limiting Treg apoptosis as a cell's survival is the result of the fine balance between proliferation and survival.

7. In Statistical analysis section on page 11 the authors stated that "experiments were repeated at least two times (n=5-10) and data are shown as a representative experiment". The figure legends stated n=3 or n=5, so I am confused whether the n=5-10 is referring to the sum of two experiments, or just a single experiment? The sample size and statistical analysis should be clearly described to help the reader understand how the conclusions were made. Due to the confusion of this, I strongly suggest the authors present their quantitative data as dot plots with the average and the appropriate error bars indicated which provides clarity and transparency and allow the audience to more critically evaluate their work. The authors should indicate how many mice were used in each groups and how many repeats in figure legends.

We apologize for the confusion and have now adapted the manuscript, Figures and Figure legends accordingly. Only single representative experiments are shown (non-pooled), and the sample size for each experiment, as well as the number of repeats, is now clearly detailed in the Figure legends, as requested. Based on each experiment, we independently repeat experiments with the same protocol and appropriate number of mice to gain confidence in our observation. The number of repeats and mice can therefore vary between experiments. This is now reflected in each figure legends to maintain accuracy, transparency and integrity. Furthermore, as requested, we have now represented our data as dot plots and added the p-value of each statistically significant result in the Figures throughout the manuscript.

Minor points

1. *On page 5 to 6, the sentence (line 111 to line 114) - As it is currently written, it feels incomplete when reading it. The authors should extend the sentence to describe the effects of the phosphorylation.*
2. *There are a few typos throughout. In the introduction, line 98, neuropilin 1 is misspelled as neutropilin 1.*
3. *Figure 1C, the y axis label, CD45 + is subscripted and should be corrected to superscript*
4. *Line 356 should be Figure 2G not Figure 1G*

We have addressed all minor comments and have adapted the manuscript and Figures accordingly, as requested.

REVIEWERS' COMMENTS

Reviewer #1 (Remarks to the Author):

The authors have responded to my initial critique with new experiments whose results address my concerns.

Reviewer #2 (Remarks to the Author):

It is important that the authors clearly describe their Tregs in each figure are from GFP+ or FOXP3+ or CD4+CD25+Foxp3+ or CD4+CD25+GFP+ cells, CD4+GFP+?.... and why?

Reviewer #3 (Remarks to the Author):

The authors carefully addressed all critical comments and significantly improve the manuscript by adding new experiments and incorporating previous references important for the field. There are no additional concerns or suggestions.

REVIEWERS' COMMENTS

We would like to thank the Editor and Reviewers for all their constructive comments. All requested experiments were reasonable and have strengthened our manuscript. Please find below the point by point answers to the Reviewers.

Reviewer #1 (Remarks to the Author):

The authors have responded to my initial critique with new experiments whose results address my concerns.

We would like to thank Reviewer #1.

Reviewer #2 (Remarks to the Author):

It is important that the authors clearly describe their Tregs in each figure are from GFP+ or FOXP3+ or CD4+CD25+Foxp3+ or CD4+CD25+GFP+ cells, CD4+GFP+?... and why?

We agree with Reviewer #2 and have added gating of Tregs when applicable in all Figure legends, in particular if Tregs were isolated from Foxp3^{GFP} mice (CD3⁺CD4⁺CD25⁺Foxp3^{GFP+}) or BALB/c mice (CD3⁺CD4⁺CD25⁺Foxp3⁺).

Reviewer #3 (Remarks to the Author):

The authors carefully addressed all critical comments and significantly improve the manuscript by adding new experiments and incorporating previous references important for the field. There are no additional concerns or suggestions.

We would like to thank Reviewer #3.